# The Effect of Conjugation of Ciprofloxacin and Moxifloxacin with Fatty Acids on Their Antibacterial and Anticancer Activity

**DOI:** 10.3390/ijms23116261

**Published:** 2022-06-02

**Authors:** Alicja Chrzanowska, Marta Struga, Piotr Roszkowski, Michał Koliński, Sebastian Kmiecik, Karolina Jałbrzykowska, Anna Zabost, Joanna Stefańska, Ewa Augustynowicz-Kopeć, Małgorzata Wrzosek, Anna Bielenica

**Affiliations:** 1Chair and Department of Biochemistry, Medical University of Warsaw, Ul. Banacha 1, 02-097 Warsaw, Poland; achrzanowska@wum.edu.pl (A.C.); mstruga@wum.edu.pl (M.S.); kjalbrzykowska@wum.edu.pl (K.J.); 2Faculty of Chemistry, University of Warsaw, Pasteura 1, 02-093 Warszawa, Poland; 3Bioinformatics Laboratory, Mossakowski Medical Research Institute, Polish Academy of Sciences, 5 Pawinskiego St., 02-106 Warsaw, Poland; kolinski.michal@gmail.com; 4Biological and Chemical Research Centre, Faculty of Chemistry, University of Warsaw, 02-089 Warsaw, Poland; sebastian.kmiecik@gmail.com; 5Department of Microbiology, National Tuberculosis and Lung Diseases Research Institute, 01-138 Warsaw, Poland; a.zabost@igichp.edu.pl (A.Z.); e.kopec@igichp.edu.pl (E.A.-K.); 6Centre for Preclinical Research, Department of Pharmaceutical Microbiology, Medical University of Warsaw, 02-097 Warszawa, Poland; jstefanska@wum.edu.pl; 7Department of Biochemistry and Pharmacogenomics, Faculty of Pharmacy, Medical University of Warsaw, 02-097 Warsaw, Poland; malgorzata.wrzosek@wum.edu.pl

**Keywords:** fluoroquinolone, conjugation, fatty acids, cytotoxicity, antibacterial activity

## Abstract

Novel conjugates (CP) of moxifloxacin (MXF) with fatty acids (**1m**–**16m**) were synthesized with good yields utilizing amides chemistry. They exhibit a more pronounced cytotoxic potential than the parent drug. They were the most effective for prostate cancer cells with an IC_50_ below 5 µM for respective conjugates with sorbic (**2m**), oleic (**4m**), 6-heptenoic (**10m**), linoleic (**11m**), caprylic (**15m**), and stearic (**16m**) acids. All derivatives were evaluated against a panel of standard and clinical bacterial strains, as well as towards mycobacteria. The highest activity towards standard isolates was observed for the acetic acid derivative **14m**, followed by conjugates of unsaturated crotonic (**1m**) and sorbic (**2m**) acids. The activity of conjugates tested against an expanded panel of clinical coagulase-negative staphylococci showed that the compound (**14m**) was recognized as a leading structure with an MIC of 0.5 μg/mL denoted for all quinolone-susceptible isolates. In the group of CP derivatives, sorbic (**2**) and geranic (**3**) acid amides exhibited the highest bactericidal potential against clinical strains. The *M. tuberculosis* Spec. 210 strain was the most sensitive to sorbic (**2m**) conjugate and to conjugates with medium- and long-chain polyunsaturated acids. To establish the mechanism of antibacterial action, selected CP and MXF conjugates were examined in both topoisomerase IV decatenation assay and the DNA gyrase supercoiling assay, followed by suitable molecular docking studies.

## 1. Introduction

Over the years, the modification of quinolones has led to changes in their chemical structure and given compounds with greater potency, pharmacokinetic properties, extended antibacterial spectrum, and less development of bacterial resistance. After the discovery of nalidixic acid (first generation), an era of further research and modification of the quinolone began [1]. Thus, in 1987, ciprofloxacin (CP) was introduced into the treatment. This second-generation fluoroquinolone is characterized by a broad spectrum of antimicrobial action with enhanced activity against Gram-positive and Gram-negative cocci. This was a consequence of subsequent modifications such as incorporation of fluorine in the R6 and a piperazine group in the R7 position (Figure 1) [2,3]. The change in the position N1 with a cyclopropyl group improved antimicrobial activity from 4 to 32 times. Further modifications initiated the next generations of fluoroquinolones, and finally the Food and Drug Administration (FDA) approved moxifloxacin (MXF) in 1999. The basic quinolone molecular structure with a fluorine atom at R6, nitrogen at N1, carboxylic acid at C3, and the ketone group at C4 remained constant, and this unchanged system ensures the maintenance of potent antibacterial activity [4]. The increase in the action on Gram-positive organisms was obtained by introducing an azabicyclic group at the R7 position, whereas adding a methoxy group at the R8 position resulted in a new component against anaerobic organisms (Figure 1) [5,6]. Both CP and MXF are recommended in the antituberculosis treatment frequency scheme as adjuncts to the basic treatment regimen [7]. Although CP shows early bactericidal activity in human tuberculosis and has been prescribed as part of a treatment for multidrug-resistant tuberculosis, the latest in vivo studies point to better efficacy of MXF [8,9].

The action of both CP and MXF is based on the inhibition of two key bacterial enzymes, DNA gyrase and topoisomerase IV [10]. These enzymes are heterotetramers. DNA gyrase is made up of the two GyrA and GyrB subunits, while topoisomerase IV is made up of two ParC and 2 ParE subunits [11]. Despite the structural similarities, DNA gyrase and topoisomerase II have different physiological functions. Analysis of mutations in gyrase, topoisomerase IV, or both enzymes in *E. coli* strains revealed gyrase as the main target for quinolones, whereas topoisomerase IV was the secondary target [12,13]. However, some reports have showed that the enzymatic target in certain bacteria depends on the structure of the respective fluoroquinolone [14,15]. Fluoroquinolones cause, by binding to the DNA/DNA-gyrase complex and inhibiting the GyrA, the inability of re-joining of the bacterial chromosome. It is known that CP is involved in the stabilization of the enzyme–DNA complex shortly after breaking the duplex strand. This results in an immediate inhibition of synthesis and the release of bacterial DNA. This process promotes the accumulation of toxic reactive oxygen species, which are responsible for the formation of cracks in bacterial chromosomes, contributing to the death of bacterial cells [16,17].

Interestingly, both CP and MXF exhibit antiproliferative and proapoptotic activity on selected cancer cell lines [18]. It is based on their affinity for bacterial-like mitochondrial DNA topoisomerase in eucaryotic cells. Beside it, anticancer activity is achieved through various mechanisms of action including mitochondrial dysfunction, oxidative stress, cell cycle arrest, influence on BCL/BCX ratio, downregulation of the levels of cyclin-CDK, and caspase activation [19]. The characteristic hallmark of some malignancies including prostate and colorectal cancers is the altered metabolism of lipids. It was established that cancer cells exhibit a modified lipid metabolism with increase of the uptake of extracellular fatty acids [20,21]. In prokaryotic cells, fatty acids are a part of numerous bacterial structures including biological membranes; therefore, the synthesis of fatty acids is essential for the survival of bacteria [22]. Fatty acids and their derivatives with good antimicrobial properties occur in nature as a part of the structure of microorganisms, algae, and plants [23]. Long-chain unsaturated fatty acids are bactericidal agents against *H. pylori*, *Mycobacterium*, and methicillin-resistant *S. aureus*. The above antibacterial properties are characteristic for oleic, linoleic, and linolenic acids. It has been established that the antimicrobial effect of saturated long-chain fatty acids is less pronounced than that of unsaturated fatty acids [24]. The strongest antibacterial properties are exhibited by dodecanoic acid (lauric, 12:0), (Z)-9-hexadecaenoic acid (palmitoleic, 16:1), and 18-carbon polyunsaturated acids. Interestingly, Gram-positive bacteria are more susceptible to the bactericidal action of fatty acids than Gram-negative rods. The esters of fatty acids and monohydroxy alcohols also have an antimicrobial effect. On the other hand, esterification of fatty acids with polyhydric alcohols (e.g., glycerol) causes an additional increase in their antibacterial activity [25]. The exact mechanism of action of long-chain fatty acids is not fully understood. It appears that the antibacterial activity of unsaturated fatty acids is related to FabI (enoyl-acyl carrier protein) inhibition. FabI is a bacterial reductase that catalyzes the final rate-limiting step in FAS II-mediated fatty acid synthesis in prokaryotic cells. Eukaryotic cells do not have a homologous structure; therefore, FabI is a potential target for antibacterial drugs [23].

Taking into account the promising results from an in vitro study on the evaluation of the cytotoxic and antibacterial effect of CP conjugates, we extended the group of fatty acids and synthetized new conjugates with a fluoroquinolone of IV generation: MXF [26]. Now, we determined the cytotoxic effect of newly synthesized conjugates on different types of cancer cells (colorectal and prostate) that exhibit active lipid metabolism, including the use of extracellular fatty acids. In addition, we evaluated their antibacterial potential as well as the possible mechanism of bactericidal action.

## 2. Results and Discussion

### 2.1. Chemistry

The objective of the research was to synthesize a series of amide derivatives of MXF by its coupling with fatty acids that differ in their chain length, degree of saturation, and position of the double bond (Figure 2). The synthetic procedure was performed under mild conditions with good yields. The characters of conjugated acyl residues were saturated (**9m**, **14m**–**16m**), as well as mono- (**1m**, **4m**, **5m**, **7m**, **10m**) or polyunsaturated (**2m**, **3m**, **6m**, **8m**, **11m**–**13m**). Among unsaturated, the group of *Z*-isomers (**4m**, **6m**–**8m**, **11m**–**13m**) was more numerous than *E*-isomers (**1m**–**3m**, **5m**). Substituents possessed short (**1m**, **2m**, **14m**), middle (**3m**, **10m**, **15m**), or long hydrocarbon chains (**4m**–**9m**, **11m**–**13m**, **16m**). The MXF-derived series is structurally comparable with a previously published group of CP analogs **1**–**9** [26].

### 2.2. Cytotoxic Activity

To establish the cytotoxic effects of MXF conjugates, they were tested for their in vitro antitumor activity on several human cancer cell lines (SW480, SW620, and PC3) *versus a* control (HaCaT) cell line. All tested cells were cultured for 72 h with different concentrations of the compounds to establish their IC_50_ value. In addition, the cytotoxic capacity against tumor cells for tested derivatives was expressed as a selectivity index (SI).

All studied derivatives were more cytotoxic against cancer than towards control HaCaT cells. The IC_50_ values determined for HaCaT cells were several dozen fold higher as compared with the studied colon cancer (SW480, SW620) and prostate (PC3) cell lines (Table 1). Analysis of cell sensitivity in individual cell lines showed the best efficiency of studied amides for the PC3 cell line. The highest cytotoxic potential (IC_50_ < 5 µM) against all studied tumor lines was shown by MXF hybrids with oleic (**4m**) and caprylic (**15m**) acids. The same level of activity towards both SW480 and PC3 cells was observed for conjugates of linoleic (**11m**) and stearic (**16m**) acids, as well as sorbic acid compound (**2m**) vs. metastatic SW620 and PC3 cell lines. Nevertheless, compounds **5m**, **8m**, and **9m** showed slightly weaker cytotoxicity (IC_50_ 5–10 µM) against all three cancer cell cultures, whereas derivatives **2m**, **10m**, **11m**, and **16m** exerted these growth-inhibitory potencies towards at least one of the tested cancerous cells. Furthermore, the above-mentioned conjugates were characterized by high selectivity factors, ranged from 11.9 to 71 (PC3 cells) and 8.7–30.2 (colon cancer cells). It is certain that both unconjugated CP and MXF had a cytotoxicity several times weaker against cancer cells than their conjugated forms, but with MXF as a stronger cytostatic than CP [26]. Most of the conjugates were more potent against SW480 and PC3 cells than the cisplatin reference. Among them, the analog **4m** was tenfold more cytotoxic towards PC3 cell lines as compared to the drug. Similarly, the compounds **2m**, **16m**, **15m**, and **11m** acted 3.2–5.5 times stronger against at least one of these cell lines. Furthermore, MXF-derived conjugates **2m**, **4m**, **10m**, **15m**, and **16m** inhibited the growth of metastatic SW620 cells more effectively than cisplatin. Although no new hybrids were as powerful as doxorubicin, all of them were incomparably more selective against malignant cells.

The correlation between the structures of MXF–fatty acids conjugates and their cytotoxic activity is imprecise. Each type of acyl residue, independently of their length and degree of unsaturation, contains both highly and weakly active analogs, as compared to the reference chemotherapeutics. The most promising derivatives, cytotoxic at concentrations below 5 µM, came from the group of middle- and long-chained molecules. They were equally represented by saturated (**15m**, **16m**), mono- (**4m**, **10m**), and di-unsaturated (**2m**, **11m**) acid amides, given mainly in Z or undefined geometrical configurations (**4m**, **10m**, **11m**). In this group, the cytotoxic properties of acyl residues decreased as follows: oleic, caprylic, stearic, linoleic, sorbic, and 6-heptenoic. The next group of highly active conjugates, effective at 10 µM towards all three cancer cells, was predominated by long-chain analogs with various degrees of unsaturation, such as elaidic (18:1), docosahexaenoic (22:6), or palmitic (16:0) acid amides. Excluding the DHA–MXF conjugate (**8m**), as the degree of unsaturation increased, the cytotoxic properties of the remaining compounds decreased. Among the less active hybrids, three of them possessed at least three double bonds in *Z* configuration (**6m**, **12m**, **13m**) or were derived from di- and monounsaturated fatty acids of various length (**3m**, **1m**, **7m**).

To compare, there is a contrast between the stronger anticancer properties of newly synthesized MXF derivatives **1m**–**9m** and CP-derived amides **1**–**9** described in our previous paper (Figure 3) [26], which were selectively active only against PC3 cells but in higher concentrations. In pairs **4**–**4m**, **2**–**2m**, and **5**–**5m** of respective CP/MXF-derived analogs, the most potent was the MXF derivative. The lowest IC_50_ values against PC3 cells reached by CP conjugates (**2**, **4**, **5**) were in the range 7.7–15.3 µM, whereas their new fluoroquinolone analogs (**2m**, **4m**, **5m**) were effective at 1.3–5.3 µM. In contrast to poorly active CP connections with DHA (**8**) and palmitic (**9**) acids (IC_50_ > 26.2 µM), their MXF-built counterparts (**8m**, **9m**) represented the group of the most cytotoxic agents towards all tested cancer cell lines, with IC_50_ values varying from 5.6 to 10.4 µM.

### 2.3. In Vitro Antibacterial Studies

The antibacterial activities of the MXF conjugates **1m**–**16m** were first evaluated against a panel of standard drug-sensitive bacterial strains, including Gram-positive (*Staphylococcus aureus, Staphylococcus epidermidis*) and Gram-negative (*Escherichia coli, Pseudomonas aeruginosa*) isolates. CP and MXF were tested under the same assay conditions as reference compounds. The minimum inhibition concentration (MIC) results were summarized in Table 2.

The highest activity towards standard Gram-positive strains was observed for amides of the short-chain fatty acids (**1m**, **2m**, and **14m**). The MIC values against staphylococci achieved by the compound **14m** varied from 0.25 μg/mL to 1 μg/mL, whereas derivatives **1m** and **2m** inhibited their growth at concentrations of 0.5–2 μg/mL. Conjugates of poly- (**8m**) and monounsaturated (**10m**) acids were moderately active (MIC 2–4 μg/mL), similarly as the saturated amide **15m** (MIC 4–8 μg/mL). Other medium- (**3m**) and long-chain (**6m**, **13m**) polyunsaturated compounds were effective in the range of 8–32 μg/mL. Long-chain amides (**4m**, **5m**, **7m**, **9m**, **11m**, **12m**, **16m**) formed a group of conjugates of the lowest antimicrobial activity (MIC > 32 μg/mL). Gram-negative isolates were more resistant to the presence of the studied compounds. Only derivatives **8m**, **14m**, **1m**, **and 13m**, applied at concentrations of 2–16 μg/mL, inhibited the growth of *E. coli* strains. Within MXF amides, the derivative **14m** was equally or even twice as active against selected *S. aureus* strains when compared to the reference CP. The compound **1m** was as potent as the standard CP towards three Staphylococcal isolates. In contrast, none of the MXF analogues were as effective as the non-conjugated drug.

The strongest potential of new derivatives was observed for *Staphylococcus* species; therefore, the activity of all conjugates was next tested against an expanded panel of clinical coagulase-negative staphylococci (CoNS). Results are given in Table 3. Characterization of antibiotic-resistant phenotypes of these bacterial strains showed that all of them are methicillin-resistant; thus, β-lactam antibiotics are excluded from therapy. Additionally, these isolates have produced other mechanisms of resistance (Appendix A). In order to characterize the profile of quinolone analogs, the studies were performed on a group of five wild-type quinolone-susceptible isolates and two quinolone-resistant species (KR 4047 and T5253). The best modification of MXF was an acetyl residue (**14m**), with an MIC of 0.5 μg/mL denoted for all quinolone-susceptible isolates. Moreover, MXF conjugates **15m** and **3m** exerted the highest activity against quinolone-resistant isolates, being so effective at concentrations of 8 μg/mL and 16–8 μg/mL, respectively, that in the case of the T 5253 strain it is eight times lower than CP alone. Both quinolones modified with crotonic (**1**, **1m**) and sorbic (**2**, **2m**) acid residues considerably inhibited the bacterial growth. The MIC values of derivatives **1m** and **2m** towards susceptible cocci reached the levels of 0.5–1 μg/mL. In the group of CP derivatives, sorbic (**2**) and geranic (**3**) acid amides were the most active. The MIC values of the compound **2** towards quinolone-susceptible isolates were at the range 0.25–0.5 μg/mL, whereas the growth inhibitory properties of the derivative **3** were observed at 1 μg/mL. It is worth noting that conjugate **2** was as effective as CP against two *S. epidermidis* isolates (KR 4243 829/19, T5399 848/19). Unsaturated linolenic acid CP-derived conjugate **6** was considerably potent against all quinolone-susceptible strains (MIC 1–2 μg/mL), whereas both heptenyl (**10m**) and arachidonyl (**13m**) MXF amides inhibited their growth at 2–8 μg/mL. Similarly, both DHA conjugates (**8**, **8m**) applied at 4 μg/mL were equally powerful towards mentioned clinical bacteria.

To sum up, the shorter the hydrocarbon chain of the quinolone derivative (**1m**, **2m**, **14m**), but not longer than six carbon atoms, the higher the antibacterial effect against standard isolates is observed. Other short-chain amides (with 7–8 carbon atoms, e.g., **10m**, **15m**), and also the most unsaturated DHA analog, predominate in the group of moderately active compounds. Medium- (**3m**) and long-chain (**6m**, **13m**) polyunsaturated derivatives were poorly active, and among them the long-chain monounsaturated (**4m**, **5m**, **7m**), di-unsaturated (**11m**, **12m**), and saturated (**9m**, **16m**) conjugates are the least potent. Elongation of the chain within monounsaturated acyl residues caused a decrease (**1m** → **10m**) or loss in bioactivity (**1m** → **4m**, **5m**, **7m**). Considering 18-carbon analogues, one can conclude that saturated (**16m**) and tri-unsaturated (**12m**) compounds are distinctly more active than their monounsaturated counterparts (**4m**, **5m**). Neither geometrical isomerism (**4m** vs. **5m**) nor an introduction of the second double bond (**4m**, **5m**
**→ 11m**) influenced considerably the antibacterial potency. The change of the position of the coupled double bonds in a pair of α-linolenic (**6m**) and γ-linolenic (**12m**) acid amides caused a 2-4-fold decrease in the growth-inhibitory activity. Comparing saturated acyl residues, the longer the aliphatic chain, the weaker activity; the antimicrobial potential diminished gradually in a group of 2-18-carbon derivatives (**14m**, **15m**, **16m**), whereas the 16-carbon molecule (**9m**) was inactive. Within conjugates having at least three double bonds, the higher the number of double bonds, the stronger the activity (**6m** → **13m** → **8m**). It was also confirmed for two pairs of counterparts: with a 22-carbon skeleton (**7m** → **8m**) and an 18-carbon chain (**4m** → **6m**). Similar observations have been made for the dependence between structures and the activity of compounds against clinical strains. The most promising group contained short-chain saturated (**14m**), mono- (**1m**), and di-unsaturated (**2**, **2m**, **3**) conjugates.

Independently from the type of the substituted quinolone, both CP-substituted [26] and MXF-derived crotonic acid amides (**1** and **1m**, respectively) were the strongest inhibitors of growth of the most standard staphylococcal strains. Considering sorbic, geranic, and α-linoleic acid amides, connection of the fatty acid with CP (**2**, **3**, **6**) was more effective than with MXF (**2m**, **3m**, **6m**). The longer the hydrocarbon chain, the higher were the differences between activities of the appropriate quinolone analogs. The DHA–MXF conjugate **8m** was an exception, being several-fold more potent than its CP analog **8**.

### 2.4. In Vitro Antimycobacterial Activity

All conjugates were studied for their in vitro preliminary antitubercular activity against the *M. tuberculosis* H_37_Rv strain and two “wild-type” mycobacteria isolated from tuberculosis patients: multidrug-resistant Spec. 210 (with resistance to p-aminosalicylic acid (PAS), INH (isoniazid), EMB (ethambutol), and RMP (rifampicin)) and Spec. 192, fully susceptible to established tuberculostatics. Commonly used antimycobacterial drugs, INH, RMP, SM, and EMB, as well as both quinolones, were used as references.

The performed studies revealed that the multidrug-resistant Spec. 210 was the most sensitive to the presence of the studied conjugates, especially those derived from medium-and long-chain polyunsaturated acids (Table 4). The sorbic acid residue combined with CP (**2**) and connections of MXF with DHA (**8m**) or arachidonic acid (**13m**) exerted 8–16-fold higher activity than referential tuberculostatics. Similarly, the DHA–CP derived compound (**8**) was 4–8 time more potent than these drugs. Unsaturated derivatives, such as amides of geranic (**3**), sorbic (**2m**), and heptenoic (**10m**) acids were equally as active as RMP and EMB, while linolenic amide **6** was as potent as INH and SM. The other two mycobacterial strains were less susceptible for quinolone derivatives. The compound **2** was the most distinctive, with MIC of 1 μg/mL being equivalently potent against Spec. 192 as both RMP and SM. The arachidonoyl amide **13m** at 4 μg/mL kept 50% of the growth-inhibitory activity of EMB towards H_37_Rv and Spec. 192 mycobacteria. Similar observations were made for the conjugate **2** against the latter strain. The antimycobacterial potential of other derivatives was considerably lower. Contrary to their antistaphylococcal properties, the group of short-chain amides (**1**, **1m**, **12m**, **14m**) was weakly active (MIC 64 μg/mL). Poor antitubercular activities (MIC ≥ 128 μg/mL) were also found for the rest of the medium- and long-chain derivatives (**3m**, **6m**, **11m**, **15m**). The bioactivity of none of the conjugates equaled the activity of the starting quinolones.

### 2.5. Inhibition of Bacterial DNA Topoisomerases

Fluoroquinolones are well-known inhibitors of bacterial topoisomerases type II (gyrase and topoisomerases IV). CP and MXF predominantly target topoisomerase IV in *S. aureus* [27], and it was shown that they probably blocked this protein through DNA binding rather than direct enzyme inhibition [28,29].

In order to examine whether the most active conjugates (**1**–**3**, **8**, **1m**–**3m**, **8m**, **14m**), similarly as parental quinolones, reveal inhibitory activity against topoisomerases, their influence on *S. aureus* DNA gyrase was studied. The synthetized conjugates, mainly the CP-derived amide **2** and **3** and MXF-derived amide **14m**, targeted gyrase (Figure 1). The half minimal inhibitory concentration (IC_50_) results (see Table 5) for their ability to inhibit DNA supercoiling by *S. aureus* gyrases were higher than reference CP and MXF and suggest that they are involved in different mechanisms of action compared to starting quinolones. In addition, these conjugates were able to inhibit the activity of bacterial topoisomerases IV from *S. aureus,* which was measured by using a decatenation assay that monitored the ATP-dependent unlinking of DNA minicircles from kDNA. We found that the topo IV decatenation reaction is also inhibited by conjugates **2**, **3**, and **14m** with an IC_50_ similar to that for supercoiling (Figure 2). As shown in Table 5, the CP-derived amide **2** targeted gyrase and topo IV to a similar extent with an IC_50_ of 34.0 ± 1.0 μg/mL and 32.0 ± 1.5 μg/mL, respectively, and had the greatest efficacy against DNA type II topoisomerases. This dual inhibition of DNA gyrase and topo IV makes compound **2** a good candidate for further drug design and development. However, moderate topo IV-inhibitory properties, especially of the most active compounds (**3**, **1m**, **14m**), could suggest that they are involved in different mechanisms of action compared to starting quinolones (Figure 2).

Decreasing amounts of compounds were incubated with 500 ng of relaxed pBR322 and run-on agarose gel. Rel is relaxed DNA; C is gyrase in the presence of a solvent; MXF is gyrase in the presence of MXF.

Lane 1: relaxed pBR322 (the omission of enzyme in the absence of drug; negative control); Lanes 2, 11, 16, 29, and 42: *S. aureus* DNA gyrase with a solvent (control). Lane: 3–6: MXF at concentrations 32, 16, 8, and 0.5 μg/mL, respectively.

Decreasing amounts of compounds were incubated with 200 ng kinetoplast DNA (kDNA) and run-on agarose gel. C is topoisomerase IV in the presence of a solvent; MXF is topoisomerase IV in the presence of MXF.

Lanes 1 and 19: *S. aureus* topoisomerase IV with a solvent (control). Lane 2: MXF at a concentration of 32 μg/mL.

### 2.6. Molecular Docking Studies

Molecular docking was used to study binding modes of the most active conjugates (**1**–**3**, **8**, **1m**–**3m**, **8m**, and **14m**) to DNA gyrase. All docked ligands preferred binding modes very similar to the binding patterns seen in complexes of DNA gyrase with DNA and CP/MXF molecules (Figure 3, Figure 4, Figure 5 and Figure 6). Namely, the aromatic rings of docked ligands were positioned and oriented similarly to these in CP and MXF in crystal structures. Moreover, in all docked ligands, the carboxyl group formed hydrogen bond/s with ARG:128 of DNA gyrase. The binding of docked ligands was also stabilized by π–π stacking interactions with TD:10 from a DNA fragment (Figure 3 and Figure 5). Likewise, the ligand–DNA gyrase/DNA interaction patterns with flanking residues were also very similar for all bound ligands.

Table 6 and Table 7 present quantitative metrics for the ranking of docked compounds: the ligand efficiency (LE) and the cluster size (CS). The ligand efficiency is the measure of binding energetics designed for comparison of ligands having different sizes. Since the binding free energy (also given in Table 6 and Table 7) is strongly biased towards the large compounds, the ligand efficiency is often used instead, which is the value of the free energy of binding divided by the number of heavy atoms in the ligand [30]. The cluster size is the measure of the docking consistency related to conformational entropy (the larger the cluster size the more consistent docking results, i.e., the larger number of similar solutions is found in different docking iterations). The combination of these two metrics is often used in virtual screening to identify the best binding molecules [30]. According to these criteria, the CP and MXF molecules can be characterized as the best binders (see Table 6 and Table 7). This is primarily because the docking consistency strongly decreases with the ligand size. Namely, 740 similar solutions were found in the CP docking results, while only 41 in the cipro-DHA (**8**) docking resulting structures. Similarly, for MXF 449 similar conformations were found, while for CS moxi-DHA (**8m**) only 44 were found. This indicates that the large size of the fatty acids substituents hinders fitting the molecules into the binding cavity. Both unconjugated compounds appeared to have favorable binding energy properties: the CP had the highest ligand efficiency among corresponding derivatives, while MXF the second highest. Furthermore, the binding energy values indicate that fatty acid fragments provide negligible or slightly attractive energetic contributions rather than repulsive (Table 6 and Table 7). The positive energetic effect is significant only in the case of DHA derivatives (**8**, **8m**). Both studies on inhibition of bacterial DNA topoisomerases and molecular docking to DNA gyrase suggest additional mechanisms of action by which the newly synthesized compounds act. Taking into account the antibacterial potential of the specific fatty acids and possible mechanism by *FabI* inhibition, it seems that this could be an additional way of antibacterial activity of fatty acid conjugates [23,31]. Interestingly, our previous proteomic study on CP conjugates and unpublished data for MXF hybrids showed their potential to reduce the human type of this enzymatic protein *Fab5* [26]. Besides, the influence of fatty acids on membrane penetration and on the phenomenon of cellular efflux may raise a hypothesis about the potential role of these processes in the antibacterial activity of the tested conjugates [32,33].

**Figure 4 ijms-23-06261-f004:**
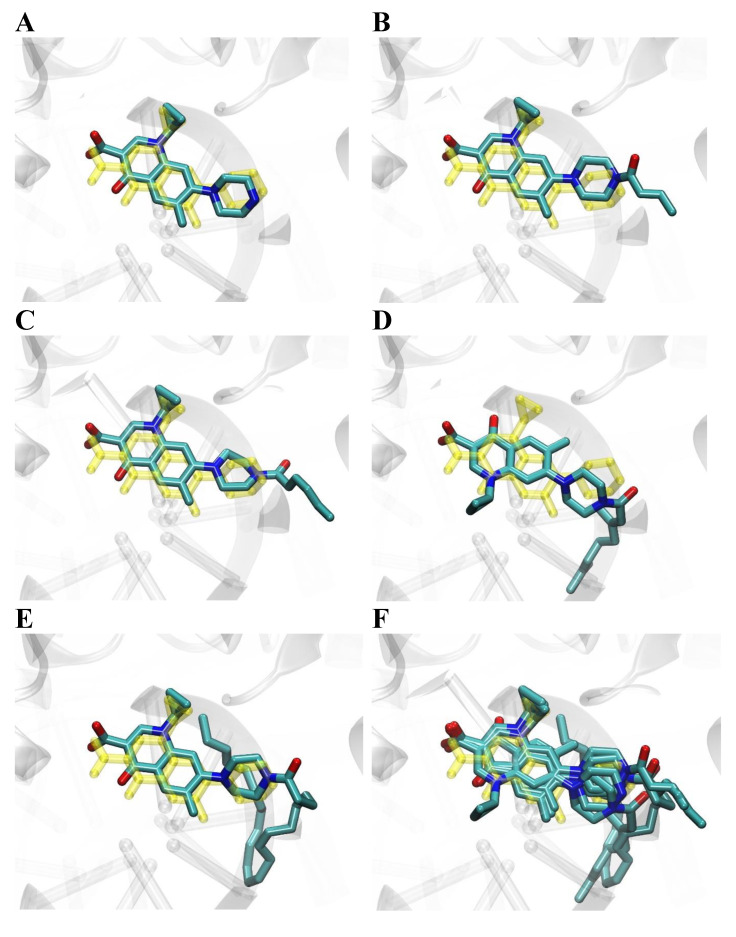
Binding modes of docked ligands to DNA gyrase structure (PDB ID: 5BTC [34]). Structure of bound CP and its four derivatives: **1**–**3** and **8** panels (**A**–**E**). Superimposition of five docked ligands is shown in panel (**F**). For comparison, the structure of CP present in the crystal structure 5BTC is shown in color yellow.

**Figure 5 ijms-23-06261-f005:**
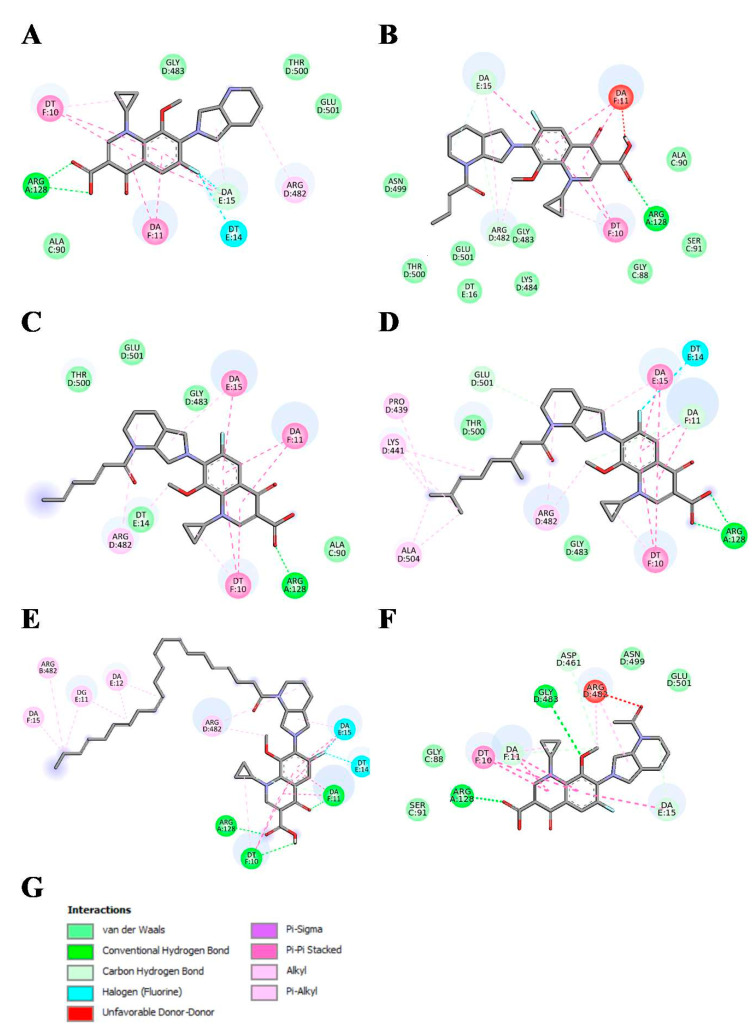
Interactions schemes generated for bound MXF, panel (**A**), and its five derivatives: **1m**–**3m**, **8m**, and **14m**, panels (**B**–**F**), respectively. Colors presenting different interaction types are shown in panel (**G**). Figure generated using BIOVIA Discovery Studio.

**Figure 6 ijms-23-06261-f006:**
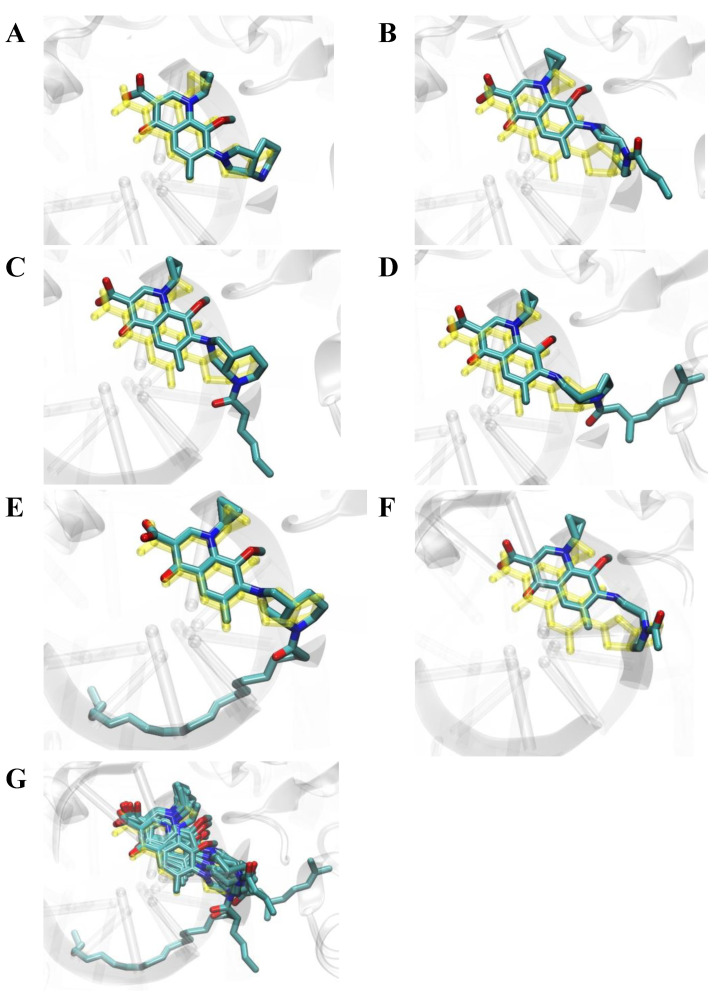
Binding modes of docked ligands to DNA gyrase structure (PDB ID: 5BS8 [34]). Structure of bound MXF and its five derivatives: **1m**–**3m**, **8m**, and **14m** panels (**A**–**F**). Superimposition of six docked ligands is shown in panel (**G**). For comparison, the structure of MXF present in the crystal structure 5BS8 is shown in color yellow.

### 2.7. Conclusions

Newly synthesized sixteen MXF–fatty acid conjugates showed high cytotoxicity against colon (SW480, SW620) and prostate (PC3) cancer cell lines but not for normal keratinocytes (HaCaT). Among the tested fatty acids related to MXF, the most effective ones were sorbic, oleic, 6-heptenoic, linoleic, caprylic, and stearic acids. The hormone-non-sensitive prostate cancer cell line PC3 appeared to be more susceptible to the action of the MXF derivatives than the primary and metastatic colon cancer cell lines. The quinolone derivatives (**1m**, **2m**, **14m**) shorter than six carbon chains expressed a higher antibacterial effect against standard isolates. Meanwhile, when it comes to clinical strains, conjugate **14m** was the strongest unit. However, both quinolones modified with crotonic (**1**, **1m**) and sorbic (**2**, **2m**) acid residues also significantly inhibited the bacterial growth. Interestingly, using CP and MXF derivatives in the antimycobacterial study showed that multidrug-resistant Spec. 210 was the most sensitive to conjugates derived from medium- and long-chain polyunsaturated acids. CP conjugate (**2**) and connections of MXF with DHA (**8m**) or arachidonic acid (**13m**) exerted 8–16-fold higher activity than referential tuberculostatics. Interestingly, after modification of both CP and MXF, the basic antibacterial mechanism of action by the inhibition of two key bacterial enzymes, DNA gyrase and topoisomerase IV, was weakened in favor of other additional ones, perhaps due to the properties of the fatty acids themselves. Therefore, we will plan further research on the potential antimicrobial mechanism of action, which will allow for a more effective indication of optimal modification and identification of new weaknesses in the bacteria metabolism that may become targets in antibiotic therapy.

## 3. Materials and Methods

### 3.1. Chemistry

Dichloromethane and methanol were supplied from Sigma-Aldrich. The *trans*-crotonic acid (98%), sorbic acid (99%), geranic acid (90+%), erucic acid, and tech. (90%) were purchased from Alfa Aesar company, DHA (98%) and CP (98%) were purchased from Acros Organics, octanoic acid (≥98%), palmitic acid (≥99%), stearic acid (95%), oleic acid (≥99%), elaidic acid (≥99%), linoleic acid (≥99%), linolenic acid (≥99%), gamma-linolenic (≥99%) acid (≥99%), arachidonic acid (>95%), and erucic acid (≥99%) were purchased from Sigma-Aldrich. All other chemicals were of analytical grade and were used without any further purification. The NMR spectra were recorded on a Bruker AVANCE spectrometer operating at 300 MHz (or 500 MHz) for ^1^H NMR and at 75 MHz (or 125 MHz) for ^13^C NMR. The spectra were measured in CDCl_3_ or DMSO-d_6_ and are given as δ values (in ppm) relative to TMS. Mass spectral ESI measurements were carried out on LCT Micromass TOF HiRes apparatus. TLC analyses were performed on silica gel plates (Merck Kiesegel GF_254_) and visualized using UV light or iodine vapor. Column chromatography was carried out at atmospheric pressure using Silica Gel 60 (230–400 mesh, Merck) and using dichloromethane/methanol (0–2%) mixture as eluent.

The synthesis of CP conjugates **1**–**9** was described previously [26].

General procedure for synthesis of MXF conjugates (**1m**–**16m**):

To a magnetically stirred cool solution (0–2 °C) of MXF (0.40 g; 0.99 mmol) and appropriate carboxylic acid (0.99 mmol) in dry CH_2_Cl_2_ (20 mL), the BOP reagent (benzotriazol-1-yloxytris(dimethylamino)phosphonium hexafluorophosphate) (0.44 g, 0.99 mmol) and triethylamine (0.21 mL; 1.49 mmol) were added. After 15 min of incubation, the cooling bath was removed, and the resulting solution was stirred at room temperature for 2 h. To the reaction mixture, dichloromethane (20 mL) and 15 mL of 3% HCl_aq_ solution were added. After phases separation, an organic layer was washed with 3% HCl_aq_ solution (3 × 15 mL) and distilled water (2 × 15 mL). Next, the organic layer was dried over anhydrous MgSO_4_. After evaporation of the solvent, the product was isolated by column chromatography on silica gel, with CH_2_Cl_2_/MeOH mixture (0–2% MeOH) as an eluent.

The ^1^H NMR and ^13^C NMR spectra of compounds **1m–16m** are given in Appendix A.
***7-{1-(E)-But-2-enoyl-octahydro-pyrrolo[3,4-b]pyridin-6-yl}-1-cyclopropyl-6-fluoro-8-methoxy-4-oxo-1,4-dihydro-quinoline-3-carboxylic acid (1m)***

Solidifying oil, 320 mg (68%).

^1^H NMR (CDCl_3_, 500 MHz) δ (ppm): 0.80–0.90 (m, 1H), 1.06–1.16 (m, 2H), 1.29–1.35 (m, 1H), 1.52–1.61 (m, 2H), 1.86–1.92 (m, 5H), 2.27–2.37 (m, 1H), 2.75–2.89 (m, 0.3H), 3.13–3.54 (m, 2.7H), 3.61 (s, 3H), 3.80–4.03 (m, 2.6H), 4.09–4.22 (m, 1H), 4.45–4.85 (m, 0.7H), 5.20–5.27 (m, 0.6H), 6.35 (d, *J* = 15.0 Hz, 1H), 6.88–6.93 (m, 1H), 7.73 (d, *J* = 14.0 Hz, 1H), 8.76 (s, 1H), 14.99 (s, 1H). ^13^C NMR (CDCl_3_, 125 MHz) δ (ppm): 8.5, 10.5, 18.3, 24.5, 25.3, 35.6, 40.4, 41.3, 48.3, 50.4, 56.4 (d, ^4^*J_C-F_* = 8.8 Hz), 61.1, 107.5, 107.8 (d, ^2^*J_C-F_* = 26.3 Hz), 118.6 (d, ^3^*J_C-F_* = 7.5 Hz), 121.8, 134.3, 137.1 (d, ^2^*J_C-F_* = 8.8 Hz), 140.9 (d, ^3^*J_C-F_* = 7.5 Hz), 142.1, 149.6, 153.5 (d, ^1^*J_C-F_* = 250.0 Hz), 166.9, 167.0, 176.6 (d, ^4^*J_C-F_* = 2.5 Hz).

HRMS (ESI) m/z 492.1925 (calculated for C_25_H_28_FN_3_O_5_Na [M+Na]^+^, 492.1911).
***1-Cyclopropyl-6-fluoro-7-{1-(2E,4E)-hexa-2,4-dienoyl-octahydro-pyrrolo[3,4-b]pyridin-6-yl}-8-methoxy-4-oxo-1,4-dihydro-quinoline-3-carboxylic acid (2m)***

Solidifying oil, 300 mg (61%).

^1^H NMR (CDCl_3_, 500 MHz) δ (ppm): 0.81–0.87 (m, 1H), 1.06–1.17 (m, 2H), 1.28–1.34 (m, 1H), 1.52–1.61 (m, 2H), 1.84–1.90 (m, 5H), 2.30–2.35 (m, 1H), 2.75–3.23 (m, 1H), 3.30–3.35 (m, 1H), 3.61 (s, 3H), 3.82–4.02 (m, 2.5H), 4.10–4.19 (m, 1H), 4.49–4.92 (m, 0.7H), 5.20–5.45 (m, 0.7H), 6.08–6.15 (m, 1H), 6.20–6.26 (m, 1H), 6.31 (d, *J* = 14.5 Hz, 1H), 7.27–7.32 (m, 1H), 7.75 (d, *J* = 13.5 Hz, 1H), 8.76 (s, 1H), 15.00 (s, 1H). ^13^C NMR (CDCl_3_, 125 MHz) δ (ppm): 8.5, 10.5, 18.6, 24.5, 25.3, 35.8, 40.4, 41.2, 48.4, 50.5, 56.4 (d, ^4^*J_C-F_* = 8.8 Hz), 61.1, 107.5, 107.8 (d, ^2^*J_C-F_* = 23.8 Hz), 117.8, 118.6 (d, ^3^*J_C-F_* = 8.8 Hz), 130.0, 134.3 (d, ^4^*J_C-F_* = 1.3 Hz), 137.1 (d, ^2^*J_C-F_* = 10.0 Hz), 138.1, 140.9 (d, ^3^*J_C-F_* = 7.5 Hz), 143.7, 149.6, 153.5 (d, ^1^*J_C-F_* = 250.0 Hz), 166.9, 167.2, 176.6 (d, ^4^*J_C-F_* = 2.5 Hz).

HRMS (ESI) m/z 518.2083 (calculated for C_27_H_30_FN_3_O_5_Na [M+Na]^+^, 518.2067).
***1-Cyclopropyl-7-{1-(2E)-(3,7-dimethyl-octa-2,6-dienoyl)-octahydro-pyrrolo[3,4-b]pyridin-6-yl}-6-fluoro-8-methoxy-4-oxo-1,4-dihydro-quinoline-3-carboxylic acid (contaminated with (Z)-isomer) (3m)***

Pale yellow oil, 300 mg (55%).

^1^H NMR (CDCl_3_, 500 MHz) δ (ppm): 0.81–0.87 (m, 1H), 1.06–1.15 (m, 2H), 1.27–1.31 (m, 1H), 1.53–1.63 (m, 2H), 1.69–1.72 (m, 6H), 1.80–1.92 (m, 5H), 2.14–2.20 (m, 4H), 2.30–2.35 (m, 1H), 2.73–2.82 (m, 0.3H), 3.10–3.19 (m, 0.7H), 3.23–3.30 (m, 1H), 3.34–3.42 (m, 0.3H), 3.52–3.56 (m, 0.7H), 3.60 (s, 3H), 3.89–4.04 (m, 2.7H), 4.08–4.15 (m, 1H), 4.60–4.71 (m, 0.7H), 5.05–5.13 (m, 1H), 5.32–5.37 (m, 0.7H), 5.77–5.84 (m, 1H), 7.75 (d, *J* = 14.0 Hz, 1H), 8.76 (s, 1H), 14.94 (s, 0.3H-R_B_), 15.01 (s, 0.7H-R_A_). ^13^C NMR (CDCl_3_, 125 MHz) δ (ppm): 8.5, 10.5, 17.7, 18.6 (2xC), 24.8 (C-R_B_), 25.4 (C-R_B_), 25.6 (C-R_A_), 25.7 (C-R_A_), 25.9, 35.5 (C-R_A_), 36.2 (C-R_B_), 39.4 (C-R_A_), 39.6 (C-R_B_), 40.4, 42.0, 48.0 (C-R_A_), 49.2 (C-R_B_), 49.8, 55.1 (C-R_B_), 56.5 (d, ^4^*J_C-F_* = 7.5 Hz, C-R_A_), 61.1, 107.5, 107.8 (d, ^2^*J_C-F_* = 22.5 Hz), 117.5 (C-R_B_), 118.0 (C-R_A_), 118.6 (d, ^3^*J_C-F_* = 8.8 Hz), 123.4 (C-R_A_), 123.6 (C-R_B_), 132.0 (C-R_B_), 132.2 (C-R_A_), 134.3, 137.2 (d, ^2^*J_C-F_* = 11.3 Hz), 141.0 (d, ^3^*J_C-F_* = 5.0 Hz), 148.3, 149.6, 153.4 (d, ^1^*J_C-F_* = 250.0 Hz, C-R_B_), 153.7 (d, ^1^*J_C-F_* = 250.0 Hz, C-R_A_), 166.9, 168.4 C-R_B_, 168.9 C-R_A_, 176.6 (d, ^4^*J_C-F_* = 3.8 Hz).

HRMS (ESI) m/z 574.2675 (calculated for C_31_H_38_FN_3_O_5_Na [M+Na]^+^, 574.2693).
***1-Cyclopropyl-6-fluoro-8-methoxy-7-{1-(Z)-octadec-9-enoyl-octahydro-pyrrolo[3,4-b] pyridin-6-yl}-4-oxo-1,4-dihydro-quinoline-3-carboxylic acid (4m)***

Pale yellow oil, 470 mg (70%).

^1^H NMR (CDCl_3_, 500 MHz) δ (ppm): 0.82–0.85 (m, 1H), 0.89 (t, *J* = 7.0 Hz, 3H), 1.06–1.18 (m, 2H), 1.29–1.35 (m, 21H), 1.53–1.59 (m, 2H), 1.64–1.68 (m, 2H), 1.84–1.92 (m, 2H), 2.01–2.04 (m, 4H), 2.26–2.38 (m, 2H), 2.41–2.55 (m, 1H), 2.74 (t, *J* = 12.0 Hz, 0.3H-R_B_), 3.19 (t, *J* = 12.0 Hz, 0.7H-R_A_), 3.27 (d, *J* = 10.5 Hz, 0.7H-R_A_), 3.34 (t, *J* = 9.5 Hz, 0.3H-R_B_), 3.40 (d, *J* = 11.5 Hz, 0.3H-R_B_), 3.52 (t, *J* = 9.5 Hz, 0.7H-R_A_), 3.60 (s, 2.1H-R_A_), 3.62 (s, 0.9H-R_B_), 3.82–3.86 (m, 1.4H), 3.99–4.03 (m, 1.4H), 4.09–4.19 (m, 1H), 4.56–4.69 (m, 0.6H-R_A_), 5.28–5.33 (m, 0.7H-R_A_), 5.36–5.38 (m, 2H), 7.80 (d, *J* = 14.0 Hz, 1H), 8.79 (s, 1H), 14.92 (s, 0.3H-R_B_), 15.01 (s, 0.7H-R_A_). ^13^C NMR (CDCl_3_, 125 MHz) δ (ppm): 8.5, 10.5, 14.1, 22.6, 23.8 (C-R_B_), 24.7 (C-R_A_), 25.2 (C-R_A_), 25.2 (C-R_A_), 25.4 (C-R_B_), 25.5 (C-R_B_), 27.1, 27.2, 29.1, 29.2, 29.3, 29.3, 29.4, 29.4, 29.6, 29.7, 31.8, 33.2 (C-R_B_), 34.1 (C-R_A_), 35.5 (C-R_A_), 36.2 (C-R_B_), 36.9 (C-R_B_), 40.3, 41.2 (C-R_A_), 48.0 (d, ^4^*J_C-F_* = 5.0 Hz, C-R_A_), 49.1 (C-R_B_), 50.0, 54.3 (C-R_B_), 56.4 (d, ^4^*J_C-F_* = 5.0 Hz, C-R_A_), 61.1, 107.6 (C-R_A_), 107.7 (C-R_B_), 107.9 (d, ^2^*J_C-F_* = 23.8 Hz, C-R_A_), 108.0 (d, ^2^*J_C-F_* = 28.7 Hz, C-R_B_), 118.7 (d, ^3^*J_C-F_* = 8.8 Hz, C-R_A_), 118.9 (d, ^3^*J_C-F_* = 10.0 Hz, C-R_B_), 129.6, 129.9, 134.3, 136.9 (d, ^2^*J_C-F_* = 10.0 Hz, C-R_B_), 137.1 (d, ^2^*J_C-F_* = 10.0 Hz, C-R_A_), 140.8 (d, ^3^*J_C-F_* = 7.5 Hz, C-R_B_), 141.0 (d, ^3^*J_C-F_* = 7.5 Hz, C-R_A_), 149.6 (C-R_A_), 149.7 (C-R_B_), 153.4 (d, ^1^*J_C-F_* = 250.0 Hz, C-R_B_), 153.7 (d, ^1^*J_C-F_* = 250.0 Hz, C-R_A_), 166.8 (C-R_B_), 166.9 (C-R_A_), 172.6 (C-R_B_), 173.2 (C-R_A_), 176.6 (d, ^4^*J_C-F_* = 3.8 Hz).

HRMS (ESI) m/z 688.4121 (calculated for C_39_H_56_FN_3_O_5_Na [M+Na]^+^, 688.4102).
***1-Cyclopropyl-6-fluoro-8-methoxy-7-{1-(E)-octadec-9-enoyl-octahydro-pyrrolo[3,4-b] pyridin-6-yl}-4-oxo-1,4-dihydro-quinoline-3-carboxylic acid (5m)***

Pale yellow oil, 500 mg (75%).

^1^H NMR (CDCl_3_, 500 MHz) δ (ppm): 0.82–0.84 (m, 1H), 0.88 (t, *J* = 7.0 Hz, 3H), 1.06–1.16 (m, 2H), 1.26–1.33 (m, 21H), 1.52–1.60 (m, 2H), 1.62–1.68 (m, 2H), 1.82–1.90 (m, 2H), 1.95–1.98 (m, 4H), 2.25–2.35 (m, 2H), 2.39–2.53 (m, 1H), 2.72 (t, *J* = 11.0 Hz, 0.3H-R_B_), 3.17 (t, *J* = 12.0 Hz, 0.7H-R_A_), 3.26 (d, *J* = 10.5 Hz, 0.7H-R_A_), 3.33 (t, *J* = 9.0 Hz, 0.3H-R_B_), 3.40 (d, *J* = 9.5 Hz, 0.3H-R_B_), 3.50 (t, *J* = 9.5 Hz, 0.7H-R_A_), 3.59 (s, 2.1H-R_A_), 3.61 (s, 0.9H-R_B_), 3.80–3.85 (m, 1.4H), 3.90–4.02 (m, 1.3H), 4.07–4.18 (m, 1H), 4.55–4.67 (m, 0.6H-R_A_), 5.26–5.31 (m, 0.7H-R_A_), 5.37–5.41 (m, 2H), 7.77 (d, *J* = 13.5 Hz, 1H), 8.77 (s, 1H), 14.92 (s, 0.3H-R_B_), 15.01 (s, 0.7H-R_A_). ^13^C NMR (CDCl_3_, 125 MHz) δ (ppm): 8.5, 10.5, 14.1, 22.6, 23.9 (C-R_B_), 24.7 (C-R_A_), 25.2 (C-R_A_), 25.2 (C-R_A_), 25.4 (C-R_B_), 25.5 (C-R_B_), 29.0, 29.1, 29.3, 29.3, 29.4, 29.4, 29.6, 29.6, 31.9, 32.5, 32.6, 33.2 (C-R_B_), 34.1 (C-R_A_), 35.5 (C-R_A_), 36.2 (C-R_B_), 36.9 (C-R_B_), 40.4, 41.2 (C-R_A_), 48.0 (d, ^4^*J_C-F_* = 6.3 Hz, C-R_A_), 49.1 (C-R_B_), 50.1, 54.4 (C-R_B_), 56.4 (d, ^4^*J_C-F_* = 6.3 Hz, C-R_A_), 61.1, 107.6 (C-R_A_), 107.6 (C-R_B_), 107.9 (d, ^2^*J_C-F_* = 23.8 Hz, C-R_A_), 108.0 (d, ^2^*J_C-F_* = 23.8 Hz, C-R_B_), 118.7 (d, ^3^*J_C-F_* = 8.8 Hz, C-R_A_), 118.8 (d, ^3^*J_C-F_* = 7.5 Hz, C-R_B_), 130.1, 130.4, 134.3, 136.9 (d, ^2^*J_C-F_* = 12.5 Hz, C-R_B_), 137.2 (d, ^2^*J_C-F_* = 10.0 Hz, C-R_A_), 140.8 (d, ^3^*J_C-F_* = 6.3 Hz, C-R_B_), 141.0 (d, ^3^*J_C-F_* = 7.5 Hz, C-R_A_), 149.6 (C-R_A_), 149.8 (C-R_B_), 153.4 (d, ^1^*J_C-F_* = 248.8 Hz, C-R_B_), 153.7 (d, ^1^*J_C-F_* = 250.0 Hz, C-R_A_), 166.8 (C-R_B_), 167.0 (C-R_A_), 172.7 (C-R_B_), 173.3 (C-R_A_), 176,3 (C-R_B_), 176.6 (d, ^4^*J_C-F_* = 2.5 Hz, C-R_A_).

HRMS (ESI) m/z 688.4125 (calculated for C_39_H_56_FN_3_O_5_Na [M+Na]^+^, 688.4102).
***1-Cyclopropyl-6-fluoro-8-methoxy-7-{1-(9Z,12Z,15Z)-octadeca-9,12,15-trienoyl-octahydro-pyrrolo[3,4-b]pyridin-6-yl}-4-oxo-1,4-dihydro-quinoline-3-carboxylic acid (6m)***

Pale yellow oil, 300 mg (45%).

^1^H NMR (CDCl_3_, 500 MHz) δ (ppm): 0.80–0.87 (m, 1H), 0.98 (t, *J* = 7.5 Hz, 3H), 1.06–1.16 (m, 2H), 1.29–1.37 (m, 9H), 1.47–1.58 (m, 2H), 1.62–1.70 (m, 2H), 1.83–1.92 (m, 2H), 1.99–2.10 (m, 4H), 2.25–2.39 (m, 2H), 2.42–2.54 (m, 1H), 2.73–2.72 (m, 4.3H), 3.18 (t, *J* = 11.5 Hz, 0.7H-R_A_), 3.27 (d, *J* = 10.0 Hz, 0.7H-R_A_), 3.38 (t, *J* = 9.5 Hz, 0.3H-R_B_), 3.39 (d, *J* = 10.5 Hz, 0.3H-R_B_), 3.50 (t, *J* = 9.5 Hz, 0.7H-R_A_), 3.59 (s, 2.1H-R_A_), 3.62 (s, 0.9H-R_B_), 3.80–3.85 (m, 1.4H), 3.98–4.01 (m, 1.3H), 4.08–4.18 (m, 1H), 4.56–4.67 (m, 0.6H-R_A_), 5.27–5.41 (m, 6.7H), 7.75 (d, *J* = 14.0 Hz, 1H), 8.76 (s, 1H), 14.93 (s, 0.3H-R_B_), 15.02 (s, 0.7H-R_A_). ^13^C NMR (CDCl_3_, 125 MHz) δ (ppm): 8.4, 10.4, 14.2, 20.5, 24.7, 25.2, 25.2, 25.4, 25.5, 27.2, 29.1, 29.3, 29.4, 29.5, 34.0, 35.4, 40.6, 41.4, 48.1, 50.1, 56.3, 61.1, 107.5, 107.8 (d, ^2^*J_C-F_* = 23.8 Hz), 118.6 (d, ^3^*J_C-F_* = 8.8 Hz, C-R_A_), 118.7 (d, ^3^*J_C-F_* = 7.5 Hz, C-R_B_), 127.0, 127.7, 128.1, 128.3, 130.0, 131.7, 134.3, 136.9 (d, ^2^*J_C-F_* = 10.0 Hz, C-R_B_), 137.2 (d, ^2^*J_C-F_* = 10.0 Hz, C-R_A_), 140.8 (d, ^3^*J_C-F_* = 6.3 Hz, C-R_B_), 141.0 (d, ^3^*J_C-F_* = 7.5 Hz, C-R_A_), 149.6 (C-R_A_), 149.7 (C-R_B_), 153.4 (d, ^1^*J_C-F_* = 250.0 Hz, C-R_B_), 153.7 (d, ^1^*J_C-F_* = 250.0 Hz, C-R_A_), 166.8 (C-R_B_), 166.9 (C-R_A_), 172.6 (C-R_B_), 173.2 (C-R_A_), 176.5 (C-R_B_), 176.6 (d, ^4^*J_C-F_* = 2.5 Hz, C-R_A_).

HRMS (ESI) m/z 684.3775 (calculated for C_39_H_52_FN_3_O_5_Na [M+Na]^+^, 684.3789).
***1-Cyclopropyl-7-{1-(Z)-docos-13-enoyl-octahydro-pyrrolo[3,4-b]pyridin-6-yl}-6-fluoro-8-methoxy-4-oxo-1,4-dihydro-quinoline-3-carboxylic acid (7m)***

Pale yellow oil, 520 mg (74%).

^1^H NMR (CDCl_3_, 500 MHz) δ (ppm): 0.82–0.84 (m, 1H), 0.88 (t, *J* = 7.0 Hz, 3H), 1.06–1.16 (m, 2H), 1.27–1.33 (m, 29H), 1.50–1.57 (m, 2H), 1.62–1.66 (m, 2H), 1.82–1.91 (m, 2H), 2.00–2.03 (m, 4H), 2.25–2.37 (m, 2H), 2.39–2.53 (m, 1H), 2.72 (t, *J* = 12.0 Hz, 0.3H-R_B_), 3.17 (t, *J* = 11.5 Hz, 0.7H-R_A_), 3.26 (d, *J* = 10.0 Hz, 0.7H-R_A_), 3.33 (t, *J* = 9.5 Hz, 0.3H-R_B_), 3.39 (d, *J* = 10.0 Hz, 0.3H-R_B_), 3.50 (t, *J* = 9.5 Hz, 0.7H-R_A_), 3.59 (s, 2.1H-R_A_), 3.61 (s, 0.9H-R_B_), 3.80–3.85 (m, 1.4H), 3.98–4.02 (m, 1.3H), 4.07–4.17 (m, 1H), 4.55–4.67 (m, 0.6H-R_A_), 5.26–5.31 (m, 0.7H-R_A_), 5.34–5.36 (m, 2H), 7.77 (d, *J* = 14.0 Hz, 1H), 8.76 (s, 1H), 14.91 (s, 0.3H-R_B_), 15.00 (s, 0.7H-R_A_). ^13^C NMR (CDCl_3_, 125 MHz) δ (ppm): 8.5, 10.5, 14.1, 22.6, 23.9 (C-R_B_), 24.7 (C-R_A_), 25.2 (C-R_A_), 25.2 (C-R_A_), 25.4 (C-R_B_), 25.5 (C-R_B_), 27.2, 29.3, 29.4, 29.4, 29.5, 29.5, 29.5, 29.6, 29.6, 29.6, 29.7, 29.7, 31.8, 33.3 (C-R_B_), 34.1 (C-R_A_), 35.5 (C-R_A_), 36.2 (C-R_B_), 36.9 (C-R_B_), 40.4, 41.2 (C-R_A_), 48.0 (d, ^4^*J_C-F_* = 6.3 Hz, C-R_A_), 49.1 (C-R_B_), 50.1, 54.3 (C-R_B_), 56.4 (d, ^4^*J_C-F_* = 6.3 Hz, C-R_A_), 61.1, 107.6 (C-R_A_), 107.6 (C-R_B_), 107.9 (d, ^2^*J_C-F_* = 23.8 Hz, C-R_A_), 108.0 (d, ^2^*J_C-F_* = 25.0 Hz, C-R_B_), 118.7 (d, ^3^*J_C-F_* = 10.0 Hz, C-R_A_), 118.8 (d, ^3^*J_C-F_* = 8.8 Hz, C-R_B_), 129.8, 129.9, 134.3, 136.9 (d, ^2^*J_C-F_* = 11.3 Hz, C-R_B_), 137.2 (d, ^2^*J_C-F_* = 11.3 Hz, C-R_A_), 140.8 (d, ^3^*J_C-F_* = 7.5 Hz, C-R_B_), 141.0 (d, ^3^*J_C-F_* = 7.5 Hz, C-R_A_), 149.6 (C-R_A_), 149.7 (C-R_B_), 153.4 (d, ^1^*J_C-F_* = 248.8 Hz, C-R_B_), 153.7 (d, ^1^*J_C-F_* = 250.0 Hz, C-R_A_), 166.8 (C-R_B_), 166.9 (C-R_A_), 172.7 (C-R_B_), 173.2 (C-R_A_), 176.6 (d, ^4^*J_C-F_* = 3.8 Hz).

HRMS (ESI) m/z 744.4711 (calculated for C_43_H_64_FN_3_O_5_Na [M+Na]^+^, 744,4728).
***1-Cyclopropyl-7-{1-(4Z,7Z,10Z,13Z,16Z,19Z)-docosa-4,7,10,13,16,19-hexaenoyl-octahydro-pyrrolo[3,4-b]pyridin-6-yl}-6-fluoro-8-methoxy-4-oxo-1,4-dihydro-quinoline-3-carboxylic acid (8m)***

Pale yellow oil, 350 mg (66%).

^1^H NMR (CDCl_3_, 500 MHz) δ (ppm): 0.80–0.88 (m, 1H), 0.97 (t, *J* = 7.5 Hz, 3H), 1.05–1.17 (m, 2H), 1.26–1.34 (m, 1H), 1.47–1.60 (m, 2H), 1.83–1.91 (m, 2H), 2.05–2.10 (m, 2H), 2.25–2.51 (m, 5H), 2.73 (t, *J* = 12.0 Hz, 0.3H-R_B_), 2.80–2.88 (m, 10H), 3.18 (t, *J* = 11.5 Hz, 0.7H-R_A_), 3.27 (d, *J* = 10.5 Hz, 0.7H-R_A_), 3.32–3.39 (m, 0.6H), 3.51 (t, *J* = 9.5 Hz, 0.7H-R_A_), 3.59 (s, 2.1H-R_A_), 3.61 (s, 0.9H-R_B_), 3.80–3.85 (m, 1.4H), 3.98–4.02 (m, 1.3H), 4.08–4.17 (m, 1H), 4.54–4.67 (m, 0.6H-R_A_), 5.26–5.46 (m, 12.7H-R_A_), 7.77 (d, *J* = 14.0 Hz, 1H), 8.77 (s, 1H), 15.00 (bs, 1H). ^13^C NMR (CDCl_3_, 125 MHz) δ (ppm): 8.5, 10.5, 14.2, 20.5, 22.6 (C-R_B_), 23.0 (C-R_A_), 23.1 (C-R_B_), 23.8 (C-R_B_), 24.7 (C-R_A_), 25.1 (C-R_A_), 25.5, 25.5, 25.6, 25.6, 25.6, 33.0 (C-R_B_), 33.7 (C-R_A_), 35.5 (C-R_A_), 36.2 (C-R_B_), 37.0 (C-R_B_), 40.4, 41.1 (C-R_A_), 48.0 (d, ^4^*J_C-F_* = 6.3 Hz, C-R_A_), 49.0 (C-R_B_), 50.2, 54.3 (C-R_B_), 56.4 (d, ^4^*J_C-F_* = 7.5 Hz, C-R_A_), 61.1, 107.5 (C-R_A_), 107.6 (C-R_B_), 107.9 (d, ^2^*J_C-F_* = 23.8 Hz, C-R_A_), 108.0 (d, ^2^*J_C-F_* = 22.5 Hz, C-R_B_), 118.7 (d, ^3^*J_C-F_* = 8.8 Hz, C-R_A_), 118.8 (d, ^3^*J_C-F_* = 8.8 Hz, C-R_B_), 126.9, 127.8, 128.0, 128.0, 128.0, 128.1, 128.2, 128.2, 128.4, 128.5, 129.0, 132.0, 134.3, 136.9 (d, ^2^*J_C-F_* = 11.3 Hz, C-R_B_), 137.2 (d, ^2^*J_C-F_* = 11.3 Hz, C-R_A_), 140.8 (d, ^3^*J_C-F_* = 6.3 Hz, C-R_B_), 141.0 (d, ^3^*J_C-F_* = 6.3 Hz, C-R_A_), 149.6 (C-R_A_), 149.8 (C-R_B_), 153.4 (d, ^1^*J_C-F_* = 250.0 Hz, C-R_B_), 153.7 (d, ^1^*J_C-F_* = 250.0 Hz, C-R_A_), 166.9 (C-R_B_), 170.0 (C-R_A_), 171.8 (C-R_B_), 172.4 (C-R_A_), 175.7 (C-R_B_), 176.6 (d, ^4^*J_C-F_* = 2.5 Hz).

HRMS (ESI) m/z 712.4149 (calculated for C_43_H_55_FN_3_O_5_ [M+H]^+^, 712.4126).
***1-Cyclopropyl-6-fluoro-7-(1-hexadecanoyl-octahydro-pyrrolo[3,4-b]pyridin-6-yl)-8-methox-4-oxo-1,4-dihydro-quinoline-3-carboxylic acid (9m)***

Solidifying oil, 485 mg (76%).

^1^H NMR (CDCl_3_, 500 MHz) δ (ppm): 0.80–0.85 (m, 1H), 0.88 (t, *J* = 6.5 Hz, 3H), 1.05–1.16 (m, 2H), 1.24–1.33 (m, 25H), 1.47–1.57 (m, 2H), 1.62–1.69 (m, 2H), 1.82–1.92 (m, 2H), 2.25–2.37 (m, 2H), 2.38–2.53 (m, 1H), 2.72 (t, *J* = 12.5 Hz, 0.3H-R_B_), 3.17 (t, *J* = 11.5 Hz, 0.7H-R_A_), 3.26 (d, *J* = 10.5 Hz, 0.7H-R_A_), 3.33 (t, *J* = 9.5 Hz, 0.3H-R_B_), 3.38 (d, *J* = 10.0 Hz, 0.3H-R_B_), 3.50 (t, *J* = 9.5 Hz, 0.7H-R_A_), 3.59 (s, 2.1H-R_A_), 3.61 (s, 0.9H-R_B_), 3.80–3.85 (m, 1.4H), 3.98–4.02 (m, 1.3H), 4.07–4.17 (m, 1H), 4.55–4.67 (m, 0.6H-R_A_), 5.26–5.31 (m, 0.7H-R_A_), 7.77 (d, *J* = 13.5 Hz, 1H), 8.77 (s, 1H), 14.92 (s, 0.3H-R_B_), 15.00 (s, 0.7H-R_A_). ^13^C NMR (CDCl_3_, 125 MHz) δ (ppm): 8.5, 10.5, 14.1, 22.6, 23.8 (C-R_B_), 24.7 (C-R_A_), 25.2 (C-R_A_), 25.2 (C-R_A_), 25.4 (C-R_B_), 25.5 (C-R_B_), 29.3, 29.4, 29.4, 29.5, 29.6, 29.6, 29.6, 29.6, 31.4, 33.2 (C-R_B_), 34.1 (C-R_A_), 35.5 (C-R_A_), 36.2 (C-R_B_), 36.9 (C-R_B_), 40.3 (C-R_A_), 40.4 (C-R_B_), 41.2 (C-R_A_), 48.0 (C-R_A_), 49.1 (C-R_B_), 50.0, 54.3 (C-R_B_), 56.4 (d, ^4^*J_C-F_* = 6.3 Hz, C-R_A_), 61.1 (C-R_A_), 61.1 (C-R_B_), 107.5 (C-R_A_), 107.6 (C-R_B_), 107.8 (d, ^2^*J_C-F_* = 23.8 Hz, C-R_A_), 108.0 (d, ^2^*J_C-F_* = 25.0 Hz, C-R_B_), 118.6 (d, ^3^*J_C-F_* = 8.8 Hz, C-R_A_), 118.8 (d, ^3^*J_C-F_* = 8.8 Hz, C-R_B_), 134.3, 136.9 (d, ^2^*J_C-F_* = 11.3 Hz, C-R_B_), 137.1 (d, ^2^*J_C-F_* = 10.0 Hz, C-R_A_), 140.8 (d, ^3^*J_C-F_* = 8.8 Hz, C-R_B_), 141.0 (d, ^3^*J_C-F_* = 7.5 Hz, C-R_A_), 149.6 (C-R_A_), 149.7 (C-R_B_), 153.4 (d, ^1^*J_C-F_* = 250.0 Hz, C-R_B_), 153.7 (d, ^1^*J_C-F_* = 250.0 Hz, C-R_A_), 166.8 (C-R_B_), 166.9 (C-R_A_), 172.7 (C-R_B_), 173.2 (C-R_A_), 176.6 (d, ^4^*J_C-F_* = 2.5 Hz).

HRMS (ESI) m/z 662.3931 (calculated for C_37_H_54_FN_3_O_5_Na [M+Na]^+^, 662.3945).
***1-Cyclopropyl-6-fluoro-7-(1-hept-6-enoyl-octahydro-pyrrolo[3,4-b]pyridin-6-yl)-8-methoxy-4-oxo-1,4-dihydro-quinoline-3-carboxylic acid (10m)***

Solidifying oil, 420 mg (82%).

^1^H NMR (CDCl_3_, 500 MHz) δ (ppm): 0.80–0.87 (m, 1H), 1.07–1.15 (m, 2H), 1.27–1.33 (m, 1H), 1.42–1.49 (m, 2H), 1.50–1.58 (m, 2H), 1.63–1.71 (m, 2H), 1.83–1.88 (m, 2H), 2.06–2.13 (m, 2H), 2.25–2.39 (m, 2H), 2.41–2.54 (m, 1H), 2.73 (t, *J* = 13.5 Hz, 0.3H-R_B_), 3.18 (t, *J* = 11.5 Hz, 0.7H-R_A_), 3.28 (d, *J* = 10.5 Hz, 0.7H-R_A_), 3.34 (t, *J* = 9.5 Hz, 0.3H-R_B_), 3.39 (d, *J* = 10.0 Hz, 0.3H-R_B_), 3.50 (t, *J* = 9.5 Hz, 0.7H-R_A_), 3.59 (s, 2.1H-R_A_), 3.62 (s, 0.9H-R_B_), 3.81–3.85 (m, 1.4H), 3.98–4.02 (m, 1.3H), 4.08–4.19 (m, 1H), 4.56–4.66 (m, 0.6H-R_A_), 4.92–5.04 (m, 2H), 5.26–5.30 (m, 0.7H-R_A_), 5.77–5.86 (m, 1H), 7.75 (d, *J* = 14.0 Hz, 1H), 8.76 (s, 1H), 14.94 (s, 0.3H-R_B_), 15.02 (s, 0.7H-R_A_). ^13^C NMR (CDCl_3_, 125 MHz) δ (ppm): 8.5, 10.5, 23.8 (C-R_B_), 24.6 (C-R_A_), 24.7 (C-R_A_), 24.8 (C-R_B_), 25.2 (C-R_A_), 25.5 (C-R_B_), 28.3 (C-R_B_), 28.6 (C-R_A_), 33.5 (C-R_A_), 33.8 (C-R_B_), 34.00 (C-R_B_), 35.5 (C-R_A_), 36.2 (C-R_B_), 36.9 (C-R_B_), 40.4, 41.2 (C-R_A_), 48.0 (d, ^4^*J_C-F_* = 6.3 Hz, C-R_A_), 49.7 (C-R_B_), 50.1 (C-R_A_), 53.4 (C-R_B_), 54.3 (C-R_B_), 56.4 (d, ^4^*J_C-F_* = 7.5 Hz, C-R_A_), 61.1, 107.5 (C-R_A_), 107.5 (C-R_B_), 107.8 (d, ^2^*J_C-F_* = 23.8 Hz, C-R_A_), 107.9 (d, ^2^*J_C-F_* = 23.8 Hz, C-R_B_), 114.4 (C-R_B_), 114.6 (C-R_A_), 118.6 (d, ^3^*J_C-F_* = 8.8 Hz, C-R_A_), 118.7 (d, ^3^*J_C-F_* = 7.5 Hz, C-R_B_), 134.3, 137.0 (d, ^2^*J_C-F_* = 12.5 Hz, C-R_B_), 137.2 (d, ^2^*J_C-F_* = 10.0 Hz, C-R_A_), 138.5, 140.8 (d, ^3^*J_C-F_* = 7.5 Hz, C-R_B_), 141.0 (d, ^3^*J_C-F_* = 7.5 Hz, C-R_A_), 149.6 (C-R_A_), 149.7 (C-R_B_), 153.4 (d, ^1^*J_C-F_* = 250.0 Hz, C-R_B_), 153.7 (d, ^1^*J_C-F_* = 248.8 Hz, C-R_A_), 166.8 (C-R_B_), 166.9 (C-R_A_), 172.5 (C-R_B_), 173.0 (C-R_A_), 175.4 (C-R_B_), 176.6 (d, ^4^*J_C-F_* = 3.8 Hz, C-R_A_).

HRMS (ESI) m/z 534.2368 (calculated for C_28_H_34_FN_3_O_5_Na [M+Na]^+^, 534.2380).
***1-Cyclopropyl-6-fluoro-8-methoxy-7-{1-(9Z,12Z)-octadeca-9,12-dienoyl-octahydropyrrolo-[3,4-b]pyridin-6-yl)-4-oxo-1,4-dihydro-quinoline-3-carboxylic acid (11m)***

Pale yellow oil, 315 mg (48%).

^1^H NMR (CDCl_3_, 500 MHz) δ (ppm): 0.80–0.85 (m, 1H), 0.89 (t, *J* = 7.0 Hz, 3H), 1.06–1.16 (m, 2H), 1.29–1.37 (m, 15H), 1.47–1.58 (m, 2H), 1.62–1.69 (m, 2H), 1.83–1.92 (m, 2H), 2.02–2.07 (m, 4H), 2.24–2.37 (m, 2H), 2.40–2.53 (m, 1H), 2.70–2.79 (m, 2.3H), 3.18 (t, *J* = 11.5 Hz, 0.7H-R_A_), 3.27 (d, *J* = 10.5 Hz, 0.7H-R_A_), 3.34 (t, *J* = 9.5 Hz, 0.3H-R_B_), 3.39 (d, *J* = 10.0 Hz, 0.3H-R_B_), 3.50 (t, *J* = 9.5 Hz, 0.7H-R_A_), 3.59 (s, 2.1H-R_A_), 3.62 (s, 0.9H-R_B_), 3.80–3.85 (m, 1.4H), 3.98–4.02 (m, 1.3H), 4.08–4.19 (m, 1H), 4.56–4.67 (m, 0.6H-R_A_), 5.27–5.41 (m, 4.7H), 7.75 (d, *J* = 13.5 Hz, 1H), 8.76 (s, 1H), 14.92 (s, 0.3H-R_B_), 15.01 (s, 0.7H-R_A_). ^13^C NMR (CDCl_3_, 125 MHz) δ (ppm): 8.5, 10.5, 14.0, 22.5, 23.9 (C-R_B_), 24.7 (C-R_A_), 25.2 (C-R_A_), 25.2 (C-R_A_), 25.4 (C-R_B_), 25.5 (C-R_B_), 25.6, 27.2, 29.1, 29.3, 29.3, 29.4, 29.6, 31.5, 33.3 (C-R_B_), 34.1 (C-R_A_), 35.5 (C-R_A_), 36.2 (C-R_B_), 36.9 (C-R_B_), 40.4, 41.2 (C-R_A_), 48.0 (d, ^4^*J_C-F_* = 6.3 Hz, C-R_A_), 49.1 (C-R_B_), 50.1, 54.3 (C-R_B_), 56.4 (d, ^4^*J_C-F_* = 7.5 Hz, C-R_A_), 61.1, 107.5 (C-R_A_), 107.6 (C-R_B_), 107.8 (d, ^2^*J_C-F_* = 23.8 Hz, C-R_A_), 107.9 (d, ^2^*J_C-F_* = 23.8 Hz, C-R_B_), 118.6 (d, ^3^*J_C-F_* = 8.8 Hz, C-R_A_), 118.7 (d, ^3^*J_C-F_* = 8.8 Hz, C-R_B_), 127.8, 128.0, 130.0, 130.2, 134.3, 136.9 (d, ^2^*J_C-F_* = 11.3 Hz, C-R_B_), 137.2 (d, ^2^*J_C-F_* = 11.3 Hz, C-R_A_), 140.8 (d, ^3^*J_C-F_* = 7.5 Hz, C-R_B_), 141.0 (d, ^3^*J_C-F_* = 7.5 Hz, C-R_A_), 149.6 (C-R_A_), 149.7 (C-R_B_), 153.4 (d, ^1^*J_C-F_* = 248.8 Hz, C-R_B_), 153.7 (d, ^1^*J_C-F_* = 250.0 Hz, C-R_A_), 166.8 (C-R_B_), 166.9 (C-R_A_), 172.6 (C-R_B_), 173.2 (C-R_A_), 176.6 (C-R_B_), 176.6 (d, ^4^*J_C-F_* = 2.5 Hz, C-R_A_).

HRMS (ESI) m/z 686.3962 (calculated for C_39_H_54_FN_3_O_5_Na [M+Na]^+^, 686.3945).
***1-Cyclopropyl-6-fluoro-8-methoxy-7-{1-(6Z,9Z,12Z)-octadeca-6,9,12-trienoyl-octahydro-pyrrolo[3,4-b]pyridin-6-yl}-4-oxo-1,4-dihydro-quinoline-3-carboxylic acid (12m)***

Pale yellow oil, 250 mg (38%).

^1^H NMR (CDCl_3_, 500 MHz) δ (ppm): 0.82–0.86 (m, 1H), 0.09 (t, *J* = 7.0 Hz, 3H), 1.05–1.17 (m, 2H), 1.29–1.38 (m, 7H), 1.42–1.48 (m, 2H), 1.54–1.60 (m, 2H), 1.65–1.71 (m, 2H), 1.83–1.91 (m, 2H), 2.04–2.08 (m, 2H), 2.10–2.14 (m, 2H), 2.23–2.38 (m, 2H), 2.41–2.55 (m, 1H), 2.72 (t, *J* = 13.0 Hz, 0.3H-R_B_), 2.79–2.82 (m, 4H), 3.18 (t, *J* = 11.5 Hz, 0.7H-R_A_), 3.27 (d, *J* = 9.5 Hz, 0.7H-R_A_), 3.36 (t, *J* = 9.5 Hz, 0.3H-R_B_), 3.39 (d, *J* = 10.0 Hz, 0.3H-R_B_), 3.50 (t, *J* = 9.5 Hz, 0.7H-R_A_), 3.59 (s, 2.1H-R_A_), 3.61 (s, 0.9H-R_B_), 3.80–3.85 (m, 1.4H), 3.98–4.02 (m, 1.3H), 4.08–4.18 (m, 1H), 4.56–4.67 (m, 0.6H-R_A_), 5.27–5.41 (m, 6.6H), 7.75 (d, *J* = 14.0 Hz, 1H), 8.76 (s, 1H), 14.94 (s, 0.3H-R_B_), 15.03 (s, 0.7H-R_A_). ^13^C NMR (CDCl_3_, 125 MHz) δ (ppm): 8.5, 10.5, 14.0, 22.4 (C-R_B_), 22.5 (C-R_A_), 23.8 (C-R_B_), 24.7 (C-R_A_), 24.8 (C-R_A_), 24.9 (C-R_B_), 25.0 (C-R_B_), 25.1 (C-R_A_), 25.6, 25.7, 26.9, 27.1, 29.2, 29.4, 31.4, 33.1 (C-R_B_), 33.9 (C-R_A_), 35.5 (C-R_A_), 36.1 (C-R_B_), 36.9 (C-R_B_), 40.4, 41.2 (C-R_A_), 48.0 (d, ^4^*J_C-F_* = 6.3 Hz, C-R_A_), 49.1 (C-R_B_), 50.1, 54.3 (C-R_B_), 56.4 (d, ^4^*J_C-F_* = 6.3 Hz, C-R_A_), 61.1, 107.5 (C-R_A_), 107.5 (C-R_B_), 107.8 (d, ^2^*J_C-F_* = 23.8 Hz), 118.6 (d, ^3^*J_C-F_* = 8.8 Hz), 127.5, 128.0, 128.1, 128.3, 129.6, 130.4, 134.3, 136.9 (d, ^2^*J_C-F_* = 11.3 Hz, C-R_B_), 137.2 (d, ^2^*J_C-F_* = 10.0 Hz, C-R_A_), 140.8 (d, ^3^*J_C-F_* = 6.3 Hz, C-R_B_), 141.0 (d, ^3^*J_C-F_* = 6.3 Hz, C-R_A_), 149.6 (C-R_A_), 149.7 (C-R_B_), 153.4 (d, ^1^*J_C-F_* = 248.8 Hz, C-R_B_), 153.7 (d, ^1^*J_C-F_* = 248.8 Hz, C-R_A_), 166.8 (C-R_B_), 167.0 (C-R_A_), 172.5 (C-R_B_), 173.0 (C-R_A_), 176.6 (C-R_B_), 176.6 (d, ^4^*J_C-F_* = 3.8 Hz, C-R_A_).

HRMS (ESI) m/z 684.3772 (calculated for C_39_H_52_FN_3_O_5_Na [M+Na]^+^, 684.3789).
***1-Cyclopropyl-6-fluoro-7-{1-(5Z,8Z,11Z,14Z)-icosa-5,8,11,14-tetraenoyl-octahydro-pyrrolo[3,4-b]pyridin-6-yl}-8-methoxy-4-oxo-1,4-dihydro-quinoline-3-carboxylic acid (13m)***

Pale yellow oil, 290 mg (42%).

^1^H NMR (CDCl_3_, 500 MHz) δ (ppm): 0.84–0.9 (m, 4H), 1.06–1.17 (m, 2H), 1.26–1.36 (m, 7H), 1.49–1.59 (m, 2H), 1.69–1.77 (m, 2H), 1.83–1.88 (m, 2H), 2.02–2.08 (m, 2H), 2.14–2.18 (m, 2H), 2.26–2.38 (m, 2H), 2.41–2.52 (m, 1H), 2.70–2.86 (m, 4.3H), 3.15–3.19 m, 0.7H-R_A_), 3.27 (d, *J* = 10.5 Hz, 0.7H-R_A_), 3.31–3.35 (m, 0.3H-R_B_), 3.36–3.39 (m, 0.3H-R_B_), 3.50 (t, *J* = 9.5 Hz, 0.7H-R_A_), 3.59 (s, 2.1H-R_A_), 3.61 (s, 0.9H-R_B_), 3.80–3.84 (m, 1.4H), 3.98–4.02 (m, 1.3H), 4.08–4.17 (m, 1H), 4.56–4.67 (m, 0.7H-R_A_), 5.27–5.41 (m, 8.6H), 7.75 (d, *J* = 14.0 Hz, 1H), 8.76 (s, 1H), 14.97 (s, 0.3H-R_B_), 15.05 (s, 0.7H-R_A_). ^13^C NMR (CDCl_3_, 125 MHz) δ (ppm): 8.5, 10.5, 14.0 (C-R_B_), 14.0 (C-R_A_), 22.4 (C-R_B_), 22.5 (C-R_A_), 23.8 (C-R_B_), 24.7 (C-R_A_), 24.9, 25.1, 25.6, 25.6, 26.7, 27.1, 29.2, 31.4, 32.4 (C-R_B_), 33.3 (C-R_A_), 35.5 (C-R_A_), 36.1 (C-R_B_), 36.9 (C-R_B_), 40.4, 41.1 (C-R_A_), 48.0 (d, ^4^*J_C-F_* = 6.3 Hz, C-R_A_), 49.0 (C-R_B_), 50.1, 54.3 (C-R_B_), 56.4 (d, ^4^*J_C-F_* = 7.5 Hz, C-R_A_), 61.1, 107.4, 107.8 (d, ^2^*J_C-F_* = 23.8 Hz), 118.6 (d, ^3^*J_C-F_* = 8.8 Hz), 127.4, 127.7, 128.1, 128.1, 128.5, 128.7, 129.1, 130.4, 134.3, 137.2 (d, ^2^*J_C-F_* = 10.0 Hz), 141.0 (d, ^3^*J_C-F_* = 7.5 Hz), 149.6 (C-R_A_), 149.7 (C-R_B_), 153.4 (d, ^1^*J_C-F_* = 246.3 Hz, C-R_B_), 153.7 (d, ^1^*J_C-F_* = 250.0 Hz, C-R_A_), 166.9 (C-R_B_), 167.0 (C-R_A_), 172.3 (C-R_B_), 172.9 (C-R_A_), 176.6 (d, ^4^*J_C-F_* = 3.8 Hz, C-R_A_).

HRMS (ESI) m/z 688.4139 (calculated for C_41_H_55_FN_3_O_5_ [M+H]^+^, 688.4126).
***7-(1-Acetyl-octahydro-pyrrolo[3,4-b]pyridin-6-yl)-1-cyclopropyl-6-fluoro-8-methoxy-4-oxo-1,4-dihydro-quinoline-3-carboxylic acid (14m)***

Solidifying oil, 340 mg (77%).

^1^H NMR (CDCl_3_, 300 MHz) δ (ppm): 0.81–0.87 (m, 1H), 1.06–1.20 (m, 2H), 1.27–1.37 (m, 1H), 1.51–1.63 (m, 2H), 1.84–1.95 (m, 2H), 2.17 (s, 2.1H-R_A_), 2.21 (s, 0.9H-R_B_), 2.25–2.43 (m, 1H), 2.73 (t, *J* = 12.3 Hz, 0.3H-R_B_), 3.20–3.31 (m, 1.3H), 3.36–3.41 (m, 0.7H), 3.51 (t, *J* = 12.3 Hz, 0.7H-R_A_), 3.61 (s, 2.1H-R_A_), 3.63 (s, 0.9H-R_B_), 3.79–3.88 (m, 1.4H), 3.97–4.03 (m, 1.4H), 4.08–4.20 (m, 1H), 4.52–4.66 (m, 0.7H-R_A_), 5.22–5.30 (m, 0.7H-R_A_), 7.72 (d, *J* = 13.8 Hz, 1H), 8.75 (s, 1H), 14.95 (s, 0.3H-R_B_), 15.03 (s, 0.7H-R_A_). ^13^C NMR (CDCl_3_, 75 MHz) δ (ppm): 8.6, 10.6, 21.7 (C-R_B_), 22.3 (C-R_A_), 23.8 (C-R_B_), 24.7 (C-R_A_), 25.2 (C-R_A_), 25.5 (C-R_B_), 35.6 (C-R_A_), 36.2 (C-R_B_), 37.0 (C-R_B_), 40.6, 42.2 (C-R_A_), 48.3 (d, ^4^*J_C-F_* = 6.0 Hz, C-R_A_), 49.1 (d, ^4^*J_C-F_* = 4.5 Hz, C-R_B_), 50.1 (C-R_A_), 53.6 (C-R_B_), 55.3 (C-R_B_), 56.5 (d, ^4^*J_C-F_* = 7.5 Hz, C-R_A_), 61.3, 107.5, 107.9 (d, ^2^*J_C-F_* = 23.7 Hz), 118.6 (d, ^3^*J_C-F_* = 9.0 Hz), 134.5, 137.3 (d, ^2^*J_C-F_* = 11.3 Hz), 141.1 (d, ^3^*J_C-F_* = 8.3 Hz), 149.7 (C-R_A_), 149.8 (C-R_B_), 153.7 (d, ^1^*J_C-F_* = 249.8 Hz), 166.9 (C-R_B_), 167.1 (C-R_A_), 170.2 (C-R_B_), 170.8 (C-R_A_), 176.7 (d, ^4^*J_C-F_* = 2.5 Hz).

HRMS (ESI) m/z 466.1742 (calculated for C_23_H_26_FN_3_O_5_Na [M+Na]^+^, 466.1754).
***1-Cyclopropyl-6-fluoro-8-methoxy-7-(1-octanoyl-octahydro-pyrrolo[3,4-b]pyridin-6-yl)-4-oxo-1,4-dihydro-quinoline-3-carboxylic acid (15m)***

Solidifying oil, 320 mg (61%).

^1^H NMR (CDCl_3_, 500 MHz) δ (ppm): 0.84–0.90 (m, 4H), 1.06–1.18 (m, 2H), 1.28–1.33 (m, 9H), 1.52–1.60 (m, 2H), 1.62–1.66 (m, 2H), 1.83–1.92 (m, 2H), 2.29–2.36 (m, 2H), 2.40–2.53 (m, 1H), 2.73 (t, *J* = 12.5 Hz, 0.3H-R_B_), 3.18 (t, *J* = 11.5 Hz, 0.7H-R_A_), 3.27 (d, *J* = 10.0 Hz, 0.7H-R_A_), 3.34 (t, *J* = 9.5 Hz, 0.3H-R_B_), 3.39 (d, *J* = 9.5 Hz, 0.3H-R_B_), 3.50 (t, *J* = 9.5 Hz, 0.7H-R_A_), 3.59 (s, 2.1H-R_A_), 3.62 (s, 0.9H-R_B_), 3.80–3.86 (m, 1.4H), 3.98–4.02 (m, 1.3H), 4.08–4.20 (m, 1H), 4.56–4.67 (m, 0.6H-R_A_), 5.26–5.31 (m, 0.7H-R_A_), 7.76 (d, *J* = 14.0 Hz, 1H), 8.77 (s, 1H), 14.94 (s, 0.3H-R_B_), 15.03 (s, 0.7H-R_A_). ^13^C NMR (CDCl_3_, 125 MHz) δ (ppm): 8.5, 10.5, 14.0 (C-R_B_), 14.0 (C-R_B_), 22.5 (C-R_B_), 22.6 (C-R_A_), 23.8 (C-R_B_), 24.7 (C-R_A_), 25.1 (C-R_A_), 25.2 (C-R_A_), 25.4 (C-R_B_), 25.5 (C-R_B_), 28.9 (C-R_B_), 29.0 (C-R_A_), 29.4 (C-R_A_), 29.4 (C-R_B_), 33.2 (C-R_B_), 34.1 (C-R_A_), 35.7 (C-R_A_), 36.2 (C-R_B_), 36.9 (C-R_B_), 40.6, 41.2 (C-R_A_), 48.0 (d, ^4^*J_C-F_* = 6.3 Hz, C-R_A_), 49.1 (C-R_B_), 50.1 (C-R_A_), 54.4 (C-R_B_), 56.4 (d, ^4^*J_C-F_* = 7.5 Hz, C-R_A_), 61.1, 107.5 (C-R_A_), 107.6 (C-R_B_), 107.8 (d, ^2^*J_C-F_* = 23.8 Hz, C-R_A_), 107.9 (d, ^2^*J_C-F_* = 23.8 Hz, C-R_B_), 118.6 (d, ^3^*J_C-F_* = 8.8 Hz, C-R_A_), 118.8 (d, ^3^*J_C-F_* = 7.5 Hz, C-R_B_), 134.3, 136.9 (d, ^2^*J_C-F_* = 10.0 Hz, C-R_B_), 137.2 (d, ^2^*J_C-F_* = 8.8 Hz, C-R_A_), 140.8 (d, ^3^*J_C-F_* = 6.3 Hz, C-R_B_), 141.0 (d, ^3^*J_C-F_* = 7.5 Hz, C-R_A_), 149.6 (C-R_A_), 149.7 (C-R_B_), 153.4 (d, ^1^*J_C-F_* = 245.0 Hz, C-R_B_), 153.7 (d, ^1^*J_C-F_* = 245.0 Hz, C-R_A_), 166.8 (C-R_B_), 167.0 (C-R_A_), 172.8 (C-R_B_), 173.3 (C-R_A_), 176.6 (d, ^4^*J_C-F_* = 2.5 Hz, C-R_A_), 177.0 (C-R_B_).

HRMS (ESI) m/z 550.2681 (calculated for C_29_H_38_FN_3_O_5_Na [M+Na]^+^, 550.2693).
***1-Cyclopropyl-6-fluoro-8-methoxy-7-(1-octadecanoyl-octahydro-pyrrolo[3,4-b]pyridin-6-yl)-4-oxo-1,4-dihydro-quinoline-3-carboxylic acid (16m)***

Solidifying oil, 480 mg (72%).

^1^H NMR (CDCl_3_, 500 MHz) δ (ppm): 0.80–0.85 (m, 1H), 0.88 (t, *J* = 7.0 Hz, 3H), 1.05–1.16 (m, 2H), 1.26–1.33 (m, 29H), 1.49–1.58 (m, 2H), 1.62–1.66 (m, 2H), 1.83–1.92 (m, 2H), 2.26–2.36 (m, 2H), 2.40–2.52 (m, 1H), 2.72 (t, *J* = 12.5 Hz, 0.3H-R_B_), 3.18 (t, *J* = 11.5 Hz, 0.7H-R_A_), 3.27 (d, *J* = 10.0 Hz, 0.7H-R_A_), 3.34 (t, *J* = 9.0 Hz, 0.3H-R_B_), 3.39 (d, *J* = 10.5 Hz, 0.3H-R_B_), 3.50 (t, *J* = 9.5 Hz, 0.7H-R_A_), 3.59 (s, 2.1H-R_A_), 3.62 (s, 0.9H-R_B_), 3.81–3.85 (m, 1.4H), 3.98–4.02 (m, 1.4H), 4.09–4.18 (m, 1H), 4.56–4.67 (m, 0.6H-R_A_), 5.26–5.31 (m, 0.7H-R_A_), 7.75 (d, *J* = 13.5 Hz, 1H), 8.76 (s, 1H), 14.92 (s, 0.3H-R_B_), 15.01 (s, 0.7H-R_A_). ^13^C NMR (CDCl_3_, 125 MHz) δ (ppm): 8.5, 10.5, 14.0, 22.6, 23.8 (C-R_B_), 24.7 (C-R_A_), 25.2 (C-R_A_), 25.2 (C-R_A_), 25.4 (C-R_B_), 25.5 (C-R_B_), 29.3, 29.4, 29.4, 29.4, 29.5, 29.6, 29.6, 31.8, 33.2 (C-R_B_), 34.0 (C-R_A_), 35.5 (C-R_A_), 36.1 (C-R_B_), 36.9 (C-R_B_), 40.3 (C-R_A_), 40.4 (C-R_B_), 41.2 (C-R_A_), 48.0 (C-R_A_), 49.1 (C-R_B_), 50.0, 54.3 (C-R_B_), 56.4 (C-R_A_), 61.1 (C-R_A_), 61.1 (C-R_B_), 107.5 (C-R_A_), 107.5 (C-R_B_), 107.8 (d, ^2^*J_C-F_* = 28.8 Hz, C-R_A_), 107.9 (d, ^2^*J_C-F_* = 27.5 Hz, C-R_B_), 118.6 (d, ^3^*J_C-F_* = 8.8 Hz, C-R_A_), 118.7 (d, ^3^*J_C-F_* = 8.8 Hz, C-R_B_), 134.3, 136.9 (d, ^2^*J_C-F_* = 11.3 Hz, C-R_B_), 137.2 (d, ^2^*J_C-F_* = 11.3 Hz, C-R_A_), 140.8 (d, ^3^*J_C-F_* = 7.5 Hz, C-R_B_), 141.0 (d, ^3^*J_C-F_* = 7.5 Hz, C-R_A_), 149.6 (C-R_A_), 149.7 (C-R_B_), 153.4 (d, ^1^*J_C-F_* = 248.8 Hz, C-R_B_), 153.7 (d, ^1^*J_C-F_* = 250.0 Hz, C-R_A_), 166.7 (C-R_B_), 166.9 (C-R_A_), 172.7 (C-R_B_), 173.2 (C-R_A_), 176.5 (C-R_B_), 176.6 (d, ^4^*J_C-F_* = 3.8 Hz, C-R_A_).

HRMS (ESI) m/z 690.4237 (calculated for C_39_H_58_FN_3_O_5_Na [M+Na]^+^, 690.4258).

### 3.2. Biological Studies

#### 3.2.1. Cell Culture

Primary and metastatic colon cancer (SW480, SW620), metastatic prostate cancer (PC3), and human immortal keratinocyte (HaCaT) cell lines were purchased from the American Type Culture Collection (Collection (ATCC, Rockville, MD, USA)) and cultured in MEM (Minimal Essential Medium, Thermo Sci, Waltham, MA, USA), RPMI 1640 (Roswell Park Memorial Institute, Biowest SAS, Nuaillé, France) and DMEM (Dulbecco’s Modified Eagle’s Medium, Biowest SAS, Nuaillé, France), respectively. Cells were seeded in 6 mL medium in a tissue culture flask (50 mL) in a 37 °C/5% CO_2_ humidified incubator. Medium was supplemented with 10% heat-inactivated fetal bovine serum (FBS), penicillin (100 U/mL), streptomycin (100 μg/mL) (Gibco BRL, San Francisco, CA, USA), and HEPES (20 mM, Thermo Sci (Waltham, MA, USA). The cells were cultured until 80–90% confluence was reached and then were harvested by treatment with 0.25% trypsin−0.02% EDTA (Gibco BRL, San Francisco, CA, USA) and used for experiments. Untreated cells were used as controls.

#### 3.2.2. MTT Assay

All MXF–fatty acid conjugates, the parental drug and leading cytostatics, doxorubicin and cisplatin, were tested at various concentrations (ranged from 5 to 140 µM). They were added on 96-well plates (1 × 10^4^ cells per well) with seeded normal and cancer cells and incubated for 72 h. MTT analysis was performed according to a previous study [26].

Cell absorbance results were inserted into the formula for the relative MTT level (%). It allows one to calculate the viability of cells under the influence of the tested compounds. Cell viability was expressed as the percentage of MTT reduction in cells treated with tested compounds compared to the control sample.

The relative MTT level was calculated using the formula: [100%] = A/B × 100% where A is the test sample absorbance, and B is the control sample absorbance. IC50 values were calculated using GraphPad Prism 6 software (GraphPad Software, San Diego, CA, USA).

#### 3.2.3. In Vitro Antibacterial Studies

To characterize antibacterial activity of quinolone conjugates, reference bacterial strains from international microbe collections—American Type Culture Collection (ATTC) and National Collection of Type Culture (NCTC), as well as a panel of *Staphylococcus epidermidis* clinical isolates—were used. The first set contains two Gram-negative organisms: *Escherichia coli* ATCC 2785 and *Pseudomonas aeruginosa* ATCC 27853, as a series of six Gram-positive strains: *S. aureus*: NCTC 4163, ATCC: 29213, 29213, 6538 and *S. epidermidis* ATCC: 12228, 35984. The clinical isolates of coagulase-negative staphylococci (CoNS) *S. epidermidis* were KR4047, KR4243, KR4268, KR4313, KR4358/2, T5253, T5399, and T5501. Antibiotic susceptibility testing including resistance phenotypes of hospital strains were determined using VITEK^®^ 2 Compact and VITEK 2 AES.

MIC (minimum inhibitory concentration) was determined by the twofold microdilution method according to the CLSI reference procedure with some modification [35]. The bacteria were cultured in brain heart infusion agar (BHI) and incubated at 37 °C for 24–48 h. Bacterial inoculum were prepared in sterile saline solution and diluted in MH II liquid medium to a final concentration of 10^6^ colony-forming units per ml (CFU/mL). The reference drugs, MXF and CP, were tested at a range 0.03125 to 32 µg/mL, whereas the concentrations of conjugates varied from 0.25 to 512 µg/mL. The final concentration of DMSO in working solutions was less than 1%. Bacteria were grown overnight in the presence of different concentrations of the tested compounds. After an 18 h period of incubation, the lowest concentration of drugs that inhibited the visible growth of bacteria were considered as the MIC value. Tests were repeated independently three times.

VITEK^®^ 2 Compact (BioMérieux) automated system for the antimicrobial susceptibility testing of microorganisms was used in accordance with the manufacturer’s directions.

#### 3.2.4. In Vitro Antimycobacterial Activity

The synthesized compounds **1**–**9** and **1m**–**16m** were tested in vitro for their tuberculostatic activity against typical strains (*M. tuberculosis* H_37_Rv strain (ATCC 25618), *M. tuberculosis* Spec. 210, *M. tuberculosis* Spec. 192) using the MABA method (microplate Alamar blue assay method) [36,37]. Investigations were performed by the twofold serial microdilution method (in 96-well microliter plates) using Middlebrook 7H9 Broth medium (Beckton Dickinson) containing 10% of OADC (Beckton Dickinson). The inoculum was prepared from fresh LJ culture in Middlebrook 7H9 Broth medium with OADC, adjusted to a no. 1 McFarland tube, and diluted 1:20. The stock solution of a tested agent was prepared in DMSO. Each stock solution of a tested compound was diluted in Middlebrook 7H9 Broth medium with OADC by fourfold the final highest concentration to be tested. Compounds were diluted serially in a sterile 96-well microtiter plate using 100 μL Middlebrook 7H9 Broth medium with OADC. Concentrations of tested agents ranged from 0.125 to 512 µg/mL. A growth control containing no antibiotic and a sterile control without inoculation were also prepared on each plate. The plates were incubated at 37 °C for 2 weeks. After the incubation period, 30μL of Alamar blue solution was added to each well, and the plate was re-incubated for 24 h. The growth was indicated by the color change from blue to pink. The lowest concentration of a compound that prevented the color change was considered as its MIC. CP, MXF, INH, RMP, SM, and EMB were used as reference drugs.

#### 3.2.5. Topoisomerases Inhibition Determination


***S. aureus* DNA Gyrase Supercoiling Assay**


The assay was performed using *S. aureus* DNA Gyrase Supercoiling kit (Inspiralis). An amount of 1 U of gyrase converts 500 ng of relaxed pBR322 DNA to the supercoiled form. Enzyme activity was detected by incubation for 30 min at 37 °C in a total reaction volume of 30 µL and in the presence of different concentrations of the test compounds. The reactions were terminated by adding an equal volume of STEB buffer (40% sucrose, 100 mM Tris-HCl pH 8, 10 mM EDTA, 0.5 mg/mL bromophenol blue), followed by extraction with 1 volume of chloroform/isoamyl alcohol (24:1). Samples were vortexed, centrifuged, and 20 µL of the aqueous phase of each sample was loaded onto a 1% agarose gel and left for 30 min prior to electrophoresis to allow diffusion of salt. Electrophoresis was conducted in TAE buffer for 3 h at 50 V. Gels were stained with ethidium bromide and visualized under UV light in a transilluminator (ChemiDoc MP, Bio Rad, Warsaw, Poland). The IC_50_ values were determined by plotting the results obtained from the densitometric analyses of the gel images using Image Lab software (BioRad, Hercules, CA, USA).
***S. aureus* Topoisomerase IV Decatenation Assay**

The assay was performed using *S. aureus* topoisomerase IV decantation kit (Inspiralis). Kinetoplast DNA (kDNA) was the substrate for topoisomerase IV. An amount of 1 U of topoisomerase IV decatenated 200 ng of kDNA, in the dedicated decantation assay buffer supplied by the manufacturer. Enzyme activity was detected by incubation for 30 min at 37 °C in a total reaction volume of 30 µL and in the presence of different concentrations of the test compounds. The reactions were terminated by adding an equal volume of STEB buffer (40% sucrose, 100 mM Tris-HCl pH 8, 1 mM EDTA, 0.5 mg/mL bromophenol blue), followed by extraction with 1 volume of chloroform/isoamyl alcohol (24: 1). Then, 20 µL of the aqueous phase of each sample was loaded onto a 1% agarose gel. Electrophoresis was conducted in TAE buffer for 1.5 h at 80 V. Gels were stained with ethidium bromide and visualized under UV light in a transilluminator (ChemiDoc MP, Bio Rad). The IC_50_ values were determined by plotting the results obtained from the densitometric analyses of the gel images using Image Lab software (BioRad).

#### 3.2.6. Molecular Docking

Molecular docking of 11 compounds to the DNA gyrase structure was performed. Docked ligands included CP and MXF, as well as their derivatives (for a list of docked ligands see Table 6 and Table 7). The structures of ligands were generated using Automated Topology Builder server (ATB version 2.2) [38]. The crystal structure PDB ID: 5BTC (DNA gyrase in complex with DNA and CP) was used for docking of CP and its derivatives, whereas the crystal structure PDB ID: 5BS8 [34] (DNA gyrase in complex with DNA and MXF) was used for docking of MXF and its derivatives. Docking calculations and data analysis were performed using AutoDock4 (v. 4.2) and AutoDockTools4 [39], respectively. For each receptor–ligand complex, 1000 independent docking cycles were performed, resulting in 1000 conformers with the lowest binding energy. Structural clustering (with RMSD cut-off of 3 Å) was then used to identify the most preferred ligand binding modes. The central structure of the largest cluster was selected as the final ligand-docked structure for each complex (Figure 3, Figure 4, Figure 5 and Figure 6).

## Data Availability

Data sharing not applicable. No new data were created or analyzed in this study. Data sharing is not applicable to this article.

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
