# Peer review of "The Effect of Conjugation of Ciprofloxacin and Moxifloxacin with Fatty Acids on Their Antibacterial and Anticancer Activity"

_ijms, 2022, doi:10.3390/ijms23116261_

Round 1
Reviewer 1 Report
The manuscript is solidly presented, methodologies of investigations are correct. The structures of new conjugates were adequately confirmed.
The manuscript is acceptable for publication after minor correcions.
My questions to the authors:
- Why did the authors choose these types of cancer? Please add some sentences to the Introduction.
- Why the authors did not conduct in silico research to topo IV but only to gyrase? Please explain this fact.
- In silico studies indicate that only docosahexaenoic acid conjugates were active. The authors should try to hypothesize what could be the reason for this fact.
- In 2.3 should be correct gram-positive and gram-negative. It should be in uppercase.
- In the Supplementary file please correct the spectrum for geranic conjugate.
Author Response
I would like to thank to the Reviewer for His/Her effort and comments, which helped improving our manuscript. The answers for the specific issues are included below. All factually changed fragments in the manuscript are marked in red.
Reviewer’s comments:
- Why did the authors choose these types of cancer? Please add some sentences to the Introduction.
The most studies conducted on cancer cells focus on glucose metabolism and the glycolytic pathway as well as on glutaminolysis. But some cancers carry a rich lipid metabolism and their progression is strongly dependent on changes in it. Due to the fact that we used different types of fatty acids for our synthesis, we chose such cell lines (colorectal and prostate cancer) which are known to carry out active lipid metabolism, including the use of external lipids. Information on the types of cell lines was supplemented in introduction (marked in red).
2. Why the authors did not conduct in silico research to topo IV but only to gyrase? Please explain this fact.
Thank you for this question. This is because the interaction pattern of moxifloxacin with both proteins is almost identical (as presented for example in figure 4B of the paper https://www.pnas.org/doi/10.1073/pnas.1525047113 ). Therefore, we focused on docking to gyrase only, since it is probably better described in the literature.
3. In silico studies indicate that only docosahexaenoic acid conjugates were active. The authors should try to hypothesize what could be the reason for this fact.
It is the most likely because of additional hydrophobic interactions of docosahexaenoic acid moiety with the protein, as presented in Figures 3 and 5.
4. In 2.3 should be correct gram-positive and gram-negative. It should be in uppercase.
It was corrected in the whole manuscript.
5. In the Supplementary file please correct the spectrum for geranic conjugate.
It was done - the spectrum was joined with the proper chemical structure.
Reviewer 2 Report
The manuscript “The effect of conjugation of ciprofloxacin and moxifloxacin with fatty acids on their antibacterial and anticancer activity” shows interesting results in terms of the activity of the new compounds. However, I would suggest the following changes before publication in “International Journal of Molecular Sciences”:
- The discussion of the activity of the compounds (sections 2.2, 2.3 and 2.4) is quite long and difficult to follow. I would suggest summarizing those sections and including only the most active compounds and the discussion of the structure-activity relationship. I would also recommend the reduction of the introduction.
- A review of the English grammar and style is required.
- I do not see the point of performing the molecular docking studies with the DNA gyrase, as it does not seem a relevant target for these compounds. Even the authors claim in the conclusion that “Both studies on inhibition of bacterial DNA topoisomerases and molecular docking to DNA gyrase clearly indicate other mechanisms of action by which the newly synthesized compounds act”.
- Most of the text in the conclusion is more appropriate for a discussion section (the discussion of other possible mechanisms of action). Please, relocate this discussion to the corresponding section.
Minor points:
1. The authors use abbreviations for ciprofloxacin (CP) and moxifloxacin (MXF) in some parts of the introduction, but not in the rest of the manuscript. The authors should be sound with the use of these abbreviations during the whole manuscript.
2. Line 187: “all of them were incomparably more selective against normal cells”. I suppose that the authors mean “more selective against malignant cells”. The sentence should be corrected accordingly.
3. Section 2.3: the affirmation “Moreover, moxifloxacin conjugates 15m and 3m exerted the highest activity against quinolone-resistant isolates, being effective at concentrations of 8 μg/ml and 16-8 μg/ml, respectively, that is 4-8 times lower than ciprofloxacin alone” is only correct for the T5253 isolate, but not for the other resistant isolate (KR4047).
4. Please include the time of treatment in tables 2, 3 and 4.
5. In section 2.4: abbreviations should be explained the first time that the authors name INH, EMB and RMP compounds, not the second one.
6. Scheme 1 is duplicated, as it appears in the manuscript and the supplementary material.
Author Response
I would like to thank to the Reviewer for His/Her effort and comments, which helped improving our manuscript. The answers for the specific issues are included below. All factually changed fragments in the manuscript are marked in red.
Reviewer’s comments:
- The discussion of the activity of the compounds (sections 2.2, 2.3 and 2.4) is quite long and difficult to follow. I would suggest summarizing those sections and including only the most active compounds and the discussion of the structure-activity relationship. I would also recommend the reduction of the introduction.
The introduction was reduced. The discussion of compounds activity in sections 2.2 and 2.3 was diminished and summarized.
- A review of the English grammar and style is required.
It was done.
- I do not see the point of performing the molecular docking studies with the DNA gyrase, as it does not seem a relevant target for these compounds. Even the authors claim in the conclusion that “Both studies on inhibition of bacterial DNA topoisomerases and molecular docking to DNA gyrase clearly indicate other mechanisms of action by which the newly synthesized compounds act”.
Indeed, this phrase in the conclusions section may be misleading – this part of Conclusion was modified and discussion of possible mechanisms of action appeared in paragraph 2.6.
- Most of the text in the conclusion is more appropriate for a discussion section (the discussion of other possible mechanisms of action). Please, relocate this discussion to the corresponding section.
The discussion of other possible mechanisms of action was moved from Conclusions to the Discussion, paragraph 2.6 (marked in red).
Minor points:
- The authors use abbreviations for ciprofloxacin (CP) and moxifloxacin (MXF) in some parts of the introduction, but not in the rest of the manuscript. The authors should be sound with the use of these abbreviations during the whole manuscript.
The abbreviations of CP and MXF were introduced to the whole manuscript.
- Line 187: “all of them were incomparably more selective against normal cells”. I suppose that the authors mean “more selective against malignant cells”. The sentence should be corrected accordingly.
The sentence was corrected.
- Section 2.3: the affirmation “Moreover, moxifloxacin conjugates 15m and 3m exerted the highest activity against quinolone-resistant isolates, being effective at concentrations of 8 μg/ml and 16-8 μg/ml, respectively, that is 4-8 times lower than ciprofloxacin alone” is only correct for the T5253 isolate, but not for the other resistant isolate (KR4047).
It was corrected as follows: “Moreover, MXF conjugates 15m and 3m exerted the highest activity against quinolone-resistant isolates, being effective at concentrations of 8 μg/ml and 16-8 μg/ml, respectively, that in case of T 5253 strain is 8 times lower than CP alone.”
- Please include the time of treatment in tables 2, 3 and 4.
The treatment times were added – 18 h (Tables 2 and 3) and 2 weeks (Table 4). The last was also clarified in the Materials and methods part.
- In section 2.4: abbreviations should be explained the first time that the authors name INH, EMB and RMP compounds, not the second one.
The corrections were made.
- Scheme 1 is duplicated, as it appears in the manuscript and the supplementary material.
The Scheme 1 was removed from Supplementary material.
Round 2
Reviewer 1 Report
The manuscript is acceptable for publication in this form.
Reviewer 2 Report
The authors have taken into account the suggestions of the reviewers and therefore, I would recommend the publication of this manuscript in Acta Tropica.